# Functional Embeddings Enable Aggregation of Multi-Area SEEG Recordings Over Subjects and Sessions

## Abstract

Aggregating intracranial recordings across subjects is challenging since electrode count, placement, and covered regions vary widely. Spatial normalization methods like MNI coordinates offer a shared anatomical reference, but often fail to capture true functional similarity, particularly when localization is imprecise; even at matched anatomical coordinates, the targeted brain region and underlying neural dynamics can differ substantially between individuals. We propose a scalable representation-learning framework that (i) learns a subject-agnostic functional identity for each electrode from multi-region local field potentials using a Siamese encoder with contrastive objectives, inducing an embedding geometry that is locality-sensitive to region-specific neural signatures, and (ii) tokenizes these embeddings for a transformer that models inter-regional relationships with a variable number of channels. We evaluate this framework on a 20-subject dataset spanning basal ganglia–thalamic regions collected during flexible rest/movement recording sessions with heterogeneous electrode layouts. The learned functional space supports accurate within-subject discrimination and forms clear, region-consistent clusters; it transfers zero-shot to unseen channels. The transformer, operating on functional tokens without subject-specific heads or supervision, captures cross-region dependencies and enables reconstruction of masked channels, providing a subject-agnostic backbone for downstream decoding. Together, these results indicate a path toward large-scale, cross-subject aggregation and pretraining for intracranial neural data where strict task structure and uniform sensor placement are unavailable.

## 1 Introduction

Building models that generalize across subjects in neuroscience requires representations that remain stable despite variability in data acquisition. Intracranial neural recordings lack this stability: electrode locations, counts, sampling, and coverage differ across individuals, reflecting clinical needs rather than standardized layouts. Without a shared representational system, cross-subject aggregation is unreliable, limiting scalable modeling and clinical translation.

These difficulties are magnified in emerging invasive multi-region datasets, where electrodes simultaneously capture activity from distinct deep-brain structures(Anonymous, 2018a; 2024a; 2023a;b;c; Cimbálník et al., 2025; Parmigiani et al., 2022; Shahabi et al., 2023; Wu et al., 2023; 2024). Such recordings are uniquely valuable for studying inter-regional communication, yet their heterogeneity makes them especially challenging to align. In practice, two obstacles dominate:

**Anatomical variability and inconsistent electrode coverage.** Electrode placement can vary significantly across subjects (Starr et al., 2006; Schönecker et al., 2015). Standard atlas-based alignment (e.g., MNI (Talairach, 1988)) assumes spatial correspondence implies functional similarity, yet recordings at matched coordinates often capture divergent functional roles(Nestor et al., 2014), and in some cases completely different brain regions (Ashkan et al., 2007; Daniluk et al., 2010).

**Multi-area recordings.** Modern DBS procedures now routinely sample from several basal ganglia and thalamic nuclei simultaneously (Anonymous, 2024b; 2018b). These datasets offer unprece-

dented opportunities to study inter-regional communication, but their heterogeneity magnifies the alignment problem: each region expresses distinct yet interdependent functional signatures.

Attempts to scale datasets, inspired by trends in language modeling (Wiggins & Tejani, 2022; Kaplan et al., 2020), are in many cases limited to multi-session data within a single subject and region (Pandarinath et al., 2018; Farshchian et al., 2018; Karpowicz et al., 2025; Degenhart et al., 2020), or combined, but through post-hoc latent alignment (Safaie et al., 2023; Ma et al., 2023), which precludes unified end-to-end training across multi-session, multi-subject datasets.

Transformer-based models have begun to address this by training foundation models for EEG decoding (Wan et al., 2023; Ding et al., 2025; Cui et al., 2024), but they typically assume fixed, high-density electrode grids and large labeled datasets. Extending these ideas to invasive recordings is harder: spatial coverage is sparse, contact locations vary across patients, and anatomical localization can be imprecise. Consequently, many approaches rely on atlas-based alignment (Mentzelopoulos et al., 2024; Chau et al., 2025), which inherits the limitations of standardized coordinate systems noted above. This motivates our central hypothesis: **Neural signals can be aligned across subjects more reliably by their functional characteristics than by their anatomical coordinates.**

We operationalize this hypothesis in *FunctionalMap*, a framework that treats "function" as the coordinate system. A Siamese encoder (Hadsell et al., 2006; Bromley et al., 1993), trained with contrastive objectives learns subject-agnostic embeddings of short neural segments, mapping signals from the same brain region close together in latent space regardless of electrode placement. These embeddings serve as functional identifiers in tokens for a transformer (Vaswani et al., 2017; Dosovitskiy et al., 2020), which models inter-regional dependencies across variable channel sets without subject IDs or subject-specific heads. To our knowledge, we are the first to learn a subject-invariant, region-structured functional coordinate system from invasive, multi-region human LFPs and to use it for end-to-end training on a unified multi-session, multi-subject dataset. **Our contributions: Functional coordinates.** We introduce data-driven embeddings that capture region-specific dynamics, yielding a subject-agnostic functional identity that replaces unreliable atlas-based alignment. **Unified cross-subject modeling.** Embedding-informed tokenization lets a single transformer aggregate heterogeneous intracranial datasets—irregular electrode layouts, multiple regions, without per-subject models. **Empirical validation.** On a 20-subject multi-region intracranial LFP dataset functional embedding was able to cluster regions and transferred zero-shot to held-out channels. when ablating coordinate systems on a 11-subject dataset, functional embeddings *significantly improve* reconstruction compared to MNI-coordinate alignment. Together, these results suggest that data-derived coordinate systems provide the missing backbone for large-scale modeling of invasive human recordings, opening the door to subject-agnostic pretraining, cross-subject decoding, and scalable representation learning in neuroscience. To make this work reproducible, all codes used in this study are available at `https://github.com/ICLR-Functional-Embedding/ICLR2026_Functional_Map`.

## 2 RELATED WORK

**Transformers for neural population modeling**   Transformer architectures have advanced modeling of neural population activity across spikes and field potentials, including NDT (Ye & Pandarinath, 2021; Ye et al., 2023), STNDT (Le & Shlizerman, 2022), and large-scale decoding frameworks (Azabou et al., 2023; Ding et al., 2025; Cui et al., 2024). These methods excel at spatiotemporal sequence modeling but generally assume stable channel identities/dense coverage or use neuronal level spiking activity, and do not provide mechanisms for aligning heterogeneous electrode sets across subjects. Our method complements these works: we retain transformer scalability while injecting *functional embeddings* that align signals across subjects and regions prior to aggregation.

**Neural decoding with SEEG**   Recent SEEG studies demonstrate decoding for speech and other behaviors from intracranial recordings (Angrick et al., 2021; Meng et al., 2021; Petrosyan et al., 2022; Wu et al., 2024). These systems are often optimized per dataset or subject and rely on task-specific supervision. In contrast, we aim to first learn subject-/region-agnostic *functional* coordinates without behavioral labels, then plug these coordinates into transformers to scale across subjects and tasks.

**Cross-subject SEEG decoding under electrode variability**  SEEG cohorts exhibit heterogeneous electrode numbers and placements across subjects, complicating multi-subject integration. Wang et al. (2023) proposes a subject and electrode agnostic neural representations using transformers trained on masked spectrogram reconstruction, our goal is not generic contextualization but functional alignment: we learn region-structured embeddings via contrastive objective, addressing the well-known gap between anatomical coordinates and functional identity. Mentzelopoulos et al. (2024) address this by tokenizing per-electrode activity, injecting 3D coordinates, and performing attention across time and electrodes with subject-specific heads. notably, Mentzelopoulos et al. (2024) found no significant performance loss when ablating positional encoding using MNI coordinates. Chau et al. (2025) pretrain a Population Transformer that aggregates arbitrary, spatially sparse channel ensembles via self-supervision, improving downstream decoding and few-shot transfer. In contrast, our approach learns *functional* embeddings directly from neural dynamics and uses them as universal coordinates for cross-subject aggregation, avoiding reliance on atlas coordinates.

## 3 Method

We propose a scalable framework for learning functionally meaningful representations from multi-region human intracranial neural recordings, designed to generalize across subjects and sessions with variable electrode configurations (Figure 1). Our approach first constructs a functional embedding space using a contrastive learning trained to map neural signals from the same brain region closer together in latent space. These learned embeddings are then used to tokenize input data and condition a transformer-based architecture for downstream tasks such as time-series reconstruction. This two-stage framework supports subject-agnostic modeling by capturing data-driven regional identity based on neural dynamics rather than anatomical location (i.e., MNI coordinates).

**Data Collection and Preprocessing**  Our dataset comprises intracranial local field potential (LFP) recordings from 20 subjects with dystonia who underwent deep brain stimulation (DBS) procedure. Stereo-EEG (SEEG) electrodes were implanted in multiple basal ganglia and thalamic nuclei, including the internal segment of the globus pallidus (GPi), the subthalamic nucleus (STN), the ventralis oralis thalamic nucleus (VO), Ventral Anterior thalamic nucleus (VA), ventral intermediate thalamic nucleus (VIM), pedunculopontine nucleus (PPN), and substantia nigra pars reticulata (SNr), with electrode targets selected based on clinical criteria. In total, the dataset includes over 442.86 electrode-hours of neural recordings. For transformer related analysis only 11 out of 20 subjects were used based on the availability of their MNI data. Further details on datasets, electrode locations, and preprocessing steps are available in Appendices A.1 and A.2.

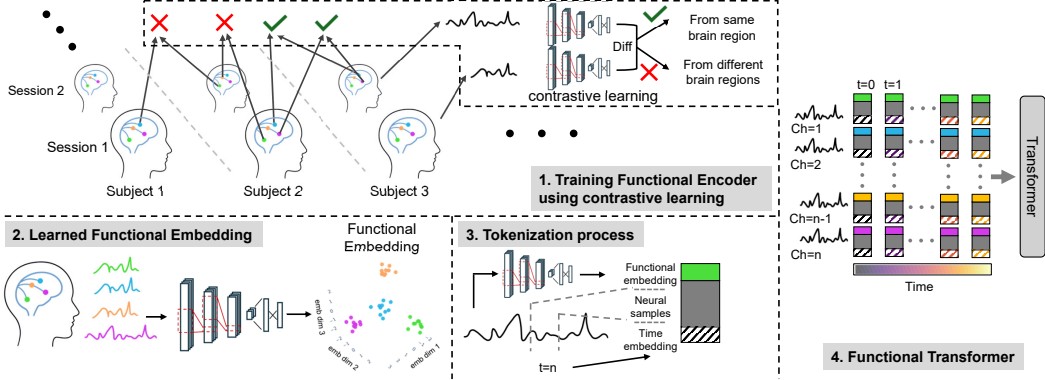

Figure 1: *Overview of Approach.* 1: Contrastive learning framework used to train the CNN-based functional encoder via a Siamese architecture, encouraging signals from the same brain region to map nearby in latent space. 2: Learned functional embeddings produced by the trained encoder, showing clustering by brain region. 3: Incorporating functional embeddings into the tokenization scheme. 4: Functional Transformer architecture that enables scalable modeling across subjects and sessions .

## 3.1 Functional Embedding Network

To address anatomical variability and heterogeneous electrode coverage across subjects, we learn a 32-dimensional (32-D) functional embedding for each recording channel using a Siamese neural network trained with contrastive loss. The objective is to construct a latent space in which signals from the same brain region, regardless of subject or session, are mapped close together (locality-sensitive functional map), thereby enabling data aggregation across sessions and individuals.

The network receives pairs/groups of LFP segments as input and minimizes a contrastive loss function that brings embeddings of positive pairs (same region) closer while pushing negative pairs (different regions) apart. The encoder architecture is a lightweight convolutional neural network (CNN) with batch normalization, GELU activation function (Hendrycks & Gimpel, 2016), and dropout (Srivastava et al., 2014). We implemented two contrastive methods:

**Method 1: Pairwise Siamese Contrastive (PSC)**
Let $f_\theta : \mathbb{R}^T \to \mathbb{R}^d$ be the encoder that maps a 10 s segment $x$ to an embedding $z = f_\theta(x)$. Given a balanced set of pairs $\{(x_i, x_j, y_{ij})\}$ with $y_{ij} \in \{0, 1\}$ (0 similar: same region; 1 dissimilar: different regions), let $z_i$ and $z_j$ be their embeddings and $d_{ij} = \|z_i - z_j\|_2$. Here, $\mathcal{B}$ denotes the set of index pairs in the current mini-batch. We minimize the standard contrastive loss with margin $m > 0$:

$$\mathcal{L}_{\text{pair}} = \frac{1}{|\mathcal{B}|} \sum_{(i,j)\in\mathcal{B}} \left[ (1 - y_{ij}) \, d_{ij}^2 + y_{ij} \left( \max(0, \, m - d_{ij}) \right)^2 \right].$$

In the case of similar pairs ($y_{ij} = 0$) only the first term in the summation is active making the loss proportional to the distance of embeddings ($d_{ij}^2$), penalizing long distances in the embedding space from same-source sample pairs. In the case of dissimilar pairs ($y_{ij} = 1$), the second term in the summation is active making the loss function proportional to the closeness of embeddings $\left( (m - d_{ij})^2 \right)$, if distance is bigger than margin ($m = 0.5$), the loss becomes zero. This encourages dissimilar pair embeddings to be at least $m$ apart(Hadsell et al., 2006).

**Method 2: Modified SupCon (MSC)**
We utilized a modified Supervised Contrastive approach (SupCon; Khosla et al. (2020)) in a single-view, projection-free setting. This method uses multi-positive InfoNCE (van den Oord et al., 2018) with label-defined positives. Let $f_\theta : \mathbb{R}^T \to \mathbb{R}^d$ be the encoder (same as PSC) producing an embedding $z_i = f_\theta(x_i)$ for segment $x_i$ with region label $y_i$. For a mini-batch $\mathcal{B} = \{(x_i, y_i)\}_{i=1}^N$, define the positive set for anchor $i$ as $\mathcal{P}(i) = \{ p \in \{1, \dots, N\} \setminus \{i\} : y_p = y_i \}$. Using inner-product similarity $s_{ij} = z_i^\top z_j$ and temperature $\tau > 0$ in our case $\tau = 0.2$:

$$\mathcal{L}_{\text{sup}} = \frac{1}{N} \sum_{i=1}^N \left( -\frac{1}{|\mathcal{P}(i)|} \sum_{p\in\mathcal{P}(i)} \log \frac{\exp(s_{ip}/\tau)}{\sum_{a\neq i} \exp(s_{ia}/\tau)} \right).$$

To further tighten same-region clusters, we add a batchwise intra-class variance term. Let $\mathcal{R}_\mathcal{B}$ be the set of region labels present in the batch and $\mathcal{B}_r = \{i \in \{1, \dots, N\} : y_i = r\}$, with class mean $\mu_r = \frac{1}{|\mathcal{B}_r|} \sum_{i\in\mathcal{B}_r} z_i$. The variance penalty is defined as:

$$\mathcal{L}_{\text{var}} = \frac{1}{|\mathcal{R}_\mathcal{B}|} \sum_{r\in\mathcal{R}_\mathcal{B}} \frac{1}{|\mathcal{B}_r|} \sum_{i\in\mathcal{B}_r} \|z_i - \mu_r\|_2^2.$$

Our final objective is

$$\mathcal{L} = \mathcal{L}_{\text{sup}} + \lambda_{\text{var}} \, \mathcal{L}_{\text{var}},$$

with $\lambda_{\text{var}} > 0$ ($\lambda_{\text{var}} = 0.05$ in our case) a tuning coefficient. No projection head and no view augmentations are used.

During training, inputs are sampled from either a single session or across sessions and subjects. In the single session setting, input pairs are not time-synchronized. In the multi-subject and session setting, pairs may originate from different individuals, sessions, and time points. This randomized, region-consistent sampling strategy ensures that the encoder learns region-specific neural signatures that are robust to subject and session variability.

**Details.** All model details and training parameters are provided in Appendix A.4, while training resources and times can be found in Appendix A.12.

## 3.2 Functional Transformer: Masked-Region Reconstruction

To directly test our hypothesis that functional data-driven embeddings capture circuit-level relationships usable across subjects, we adopt masked-region reconstruction: withhold all channels from one region and require the model to predict them from the remaining regions. This task is label-free, runs on any multi-region dataset, isolates purely neural-circuit information, and naturally serves as a self-supervised pretraining objective for downstream decoding. For each window, multichannel LFP $\mathbf{X} \in \mathbb{R}^{C \times T}$ is split into *sources* $\mathcal{S}$ (unmasked) and *targets* $\mathcal{T}$ (masked) channels, and each channel has a 32-D functional embedding extracted once from a 10 s snippet.

**Tokenization.** A 1D conv tokenizer converts each source channel into $P$ time-patch features (width $d$). The channel's functional embedding is mapped to width $d$, broadcast across its patches, concatenated with the conv features, reduced back to $d$, and augmented with a time-only positional code to form *source tokens*. Targets have no signal; we build *query tokens* by fusing a learned per-patch query base with the target's functional embedding and the same positional code. Variable numbers of sources/targets are batched with padding masks.

**Architecture.** A standard pre-LN encoder–decoder Transformer operates on these sequences: the encoder applies self-attention over source tokens to produce a memory; the decoder applies self-attention over queries followed by cross-attention to the memory, then a linear head maps each decoder token to a waveform patch that we concatenate to form the reconstruction. No subject IDs, subject-specific heads, or per-subject fine-tuning are used; a single shared model is trained across all subjects.

**Objective.** We optimize a mixed loss that combines sample-wise MSE with a correlation term to emphasize waveform shape:

$$\mathcal{L} = \mathrm{MSE}(\widehat{\mathbf{Y}}, \mathbf{Y}) \, + \, \lambda\big(1 - \rho(\widehat{\mathbf{Y}}, \mathbf{Y})\big), \quad \lambda = 0.05,$$

where $\rho$ is mean Pearson correlation across targets. The correlation term counteracts the tendency of pure MSE to favor amplitude shrinkage/flat predictions under scale or SNR mismatch.

**Details.** All token shapes, attention formulas (MHSA/MHA with Q/K/V sources), and masking/batching mechanics, model details and training parameters are provided in Appendix A.5. Training resources and times can be found in Appendix A.12

## 4 Results

### 4.1 Functional Embeddings reveal neural signature of each brain region across subjects (simulation study)

To assess whether the functional embedding network captures physiologically meaningful neural signatures, we designed a series of interpretability analyses using both simulated and analytic approaches.

**Simulation-based validation with ground-truth neural signatures.** We first generated a simulated dataset in which each brain region was assigned a parameterized neural signature, primarily defined in the spectral domain, and recordings were synthesized across 10 subjects and 2 sessions (additional details in Appendix A.3). Subject/session variability was introduced by scaling, frequency shifts, and noise (Fig. A.2). Training on a subset of subjects and evaluating on held-out subjects, the functional embedding network trained with contrastive learning (PSC) correctly clustered recordings by region with high accuracy (Fig. 2A). This demonstrates that the embedding framework is capable of extracting stable, region-specific neural signatures even in the presence of cross-subject variability.

**Perturbation-based attribution.** We next probed which input features (sub-windows) contributed most strongly to embedding location. Short temporal segments of the input were selectively frozen while other parts of the signal were perturbed, and the resulting displacement in embedding space was quantified (saliency map). Freezing segments overlapping the bursts that defined the region's neural signature produced the smallest embedding shifts (Fig. 2B bottom). This confirms that the embeddings are sensitive to the physiologically relevant components of the signal.

**Spectral properties of embedding centroids.** To further link the embedding space to interpretable features, we computed centroids for each region cluster and examined their spectral properties. The power spectral densities of signals nearest to each centroid revealed frequency characteristics that

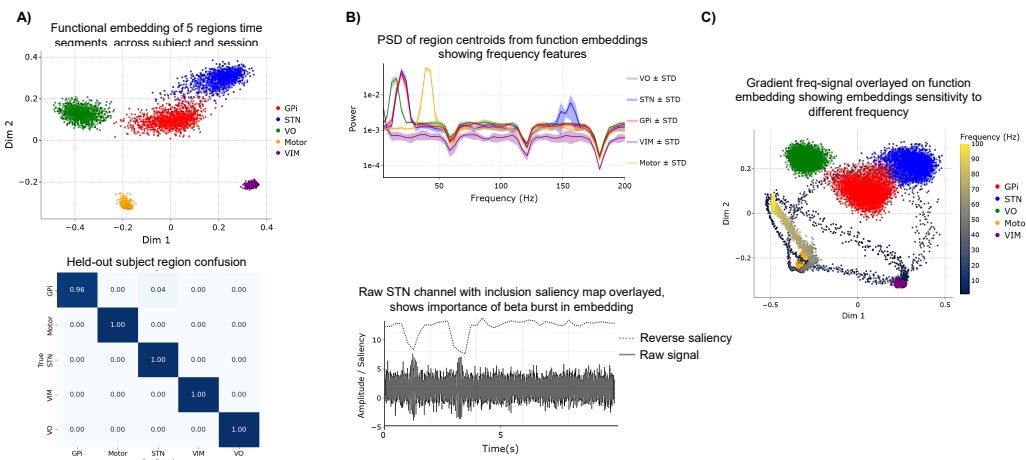

Figure 2: *Simulation-based validation of the functional embedding.* **(A)** Contrastive encoder trained on simulated data clusters test segments by region and generalizes to held-out subjects (scatter, top) with high accuracy (confusion matrix, bottom). **(B)** Top: Power spectra of signals nearest each region centroid match the ground-truth signatures; Bottom: Perturbation saliency shows largest embedding shifts when disrupting signature bursts. **(C)** Smooth frequency sweeps trace continuous trajectories in the embedding, passing near region centroids tuned to corresponding bands, indicating a locality-sensitive functional map.

matched the region's simulated neural signature (Fig. 2B top). This shows that the embedding clusters correspond to distinct and physiologically interpretable frequency-domain patterns.

**Smoothness of the functional map.** Finally, we evaluated the continuity of the embedding space in encoding signal properties, by presenting pure sinusoidal inputs with gradually increasing frequency. The embeddings formed smooth trajectories through the functional space, passing near centroids of regions tuned to corresponding frequency bands (Fig. 2C). This demonstrates that the functional embedding defines a continuous map where novel signals can be meaningfully placed relative to known regions.

### 4.2 FUNCTIONAL EMBEDDING LEARNED WITHIN INDIVIDUAL SUBJECTS PROVIDE A LOCALITY SENSITIVE EMBEDDING SPACE ENCODING FUNCTION

Across 20 subjects (most with multiple sessions), per-subject functional encoders (using PSC) trained on 10 s segments achieved a mean test accuracy of $75.78\% \pm 17.90\%$ (M±SD) on held-out time segments from electrodes seen during training (Fig. 3A, top). Under the stricter held-out channel evaluation (regions with $> 3$ channels), accuracy remained above chance at $45.79\% \pm 18.44\%$ across subjects, with some reaching $\sim 70\%$ (Fig. 3A, bottom). For an example subject (S4), the 32-D embeddings projected via PCA exhibit clear region-specific separation for held-out time segments, reflecting a locality-preserving organization of the learned space (Fig. 3B, bottom). Applying $k$-NN on these embeddings yields confusion matrices that are strongly diagonal for held-out time segments and noticeably weaker for held-out channels in this subject (Fig. 3B, top). The results for other subjects are included in Appendix A.11. Overall, subject-specific functional encoders demonstrate reliable within-subject, region-sensitive representations; channel-level generalization is lower than time-segment generalization, likely due to the limited number of electrodes per region within a subject ($\sim 6$–12), yet on average remains above chance.

### 4.3 AGGREGATED DATA OVER SUBJECTS ENCODE BRAIN REGIONS

**Functional encoder successfully learns combined dataset.** When the functional embedding was trained jointly on the entire dataset—sampling similar/dissimilar pairs across subjects and sessions (using PSC), it learned a global, subject-independent mapping with region-specific neural signatures. On unseen time segments, it achieved $80.71\% \pm 11.41\%$ (M±SD) accuracy. The same network generalized zero-shot to held-out channels, yielding $49.18\% \pm 12.11\%$ accuracy (above

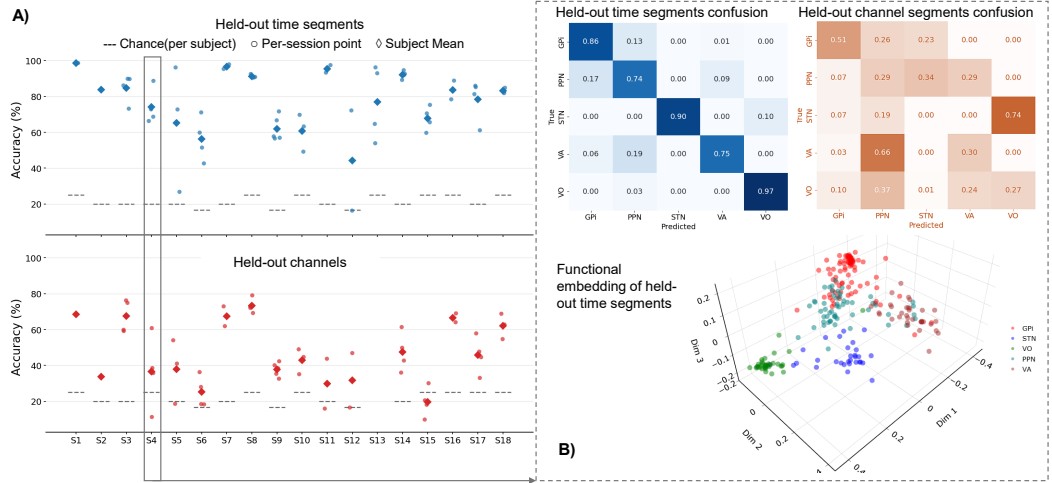

Figure 3: *Single-subject functional embeddings.* **(A)** Per-subject accuracies: top—held-out time segments from seen channels; bottom—held-out channels (only regions with $> 3$ channels). Diamonds denote subject means across sessions; circles denote session-level points; dashed lines indicate per-subject chance levels. **(B)** Example subject (S4): top — confusion matrices from $k$-NN on embeddings for held-out time segments (left) and held-out channels (right); bottom — 3D PCA visualization of the 32-D embedding space, color-coded by brain region.

per-subject chance). Compared to training a separate CNN per subject, the joint model performed better by $\sim 5\%$ on average for both held-out time segments and held-out channels without any subject-specific fine-tuning, demonstrating successful aggregation across 20 subjects in a unified model. A head-to-head comparison per recording session shows the multi-subject model surpasses single-subject models in most cases, especially where single-subject performance was low, likely due to limited channels per region (Fig. 4). The performance variability across subjects also decreased under the multi-subject model, consistent with a more robust and generalizable encoder.

**Contrastive objective comparison.** We compared two previously mentioned contrastive methods trained under the same multi-subject regime (Fig. 5). Both PSC and MSC created embedding spaces that segregates brain regions and clusters samples from same region close together. PSC yields slightly higher accuracy on held-out time segments, whereas MSC clearly outperforms on held-out channels, indicating stronger channel-robust generalization (5). The embedding geometries differ: MSC computes dot products between $\ell_2$-normalized vectors (cosine similarity), which constrains representations to a hypersphere in the 32-D space and emphasizes angular separation; PSC operates in Euclidean feature space, producing more compact, centroid-like clusters. Both embeddings are tested as tokenizers for the transformer.

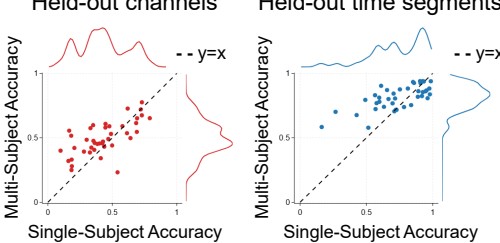

Figure 4: *Cross-subject aggregation* Head-to-head comparison of the multi-subject model (y-axis) versus single-subject models (x-axis) for held-out time (left) and held-out channels (right). Each point is a recording session; the dashed line denotes $y = x$. Marginal densities summarize performance distributions.

## 4.4 FUNCTIONAL EMBEDDINGS ENABLE AGGREGATION OF NEURAL DATA OVER SUBJECTS USING TRANSFORMERS

We tested whether functional embeddings provide a subject-agnostic coordinate system that improves cross-subject pooling for masked-region reconstruction. In each evaluation window, all channels from a target region (here, VO) were withheld, signals from the remaining regions (i.e., GPi, STN, and VIM) were encoded as source tokens, and the model queried the withheld VO chan-

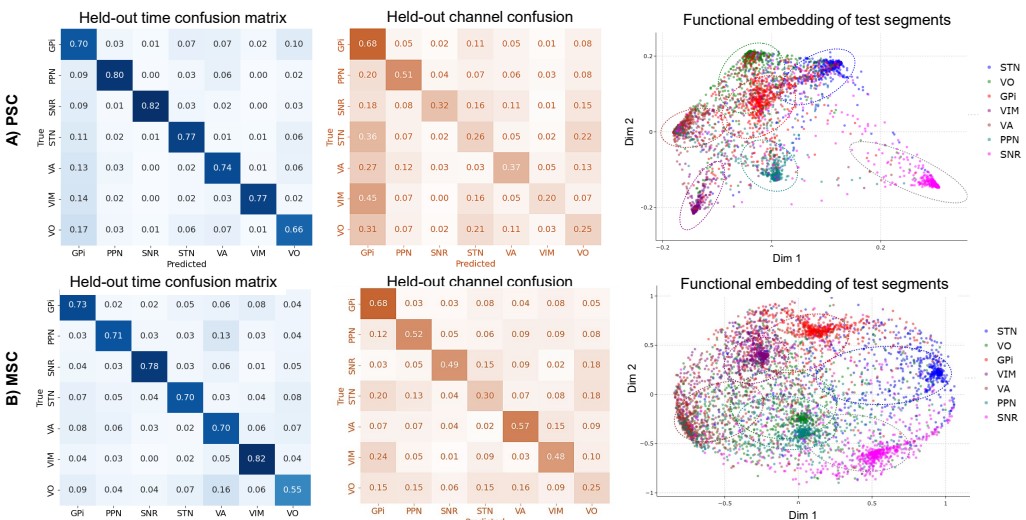

Figure 5: *Comparison of contrastive Methods (PSC vs. MSC)* **A) PSC:** Row-normalized confusion matrices from $k$-NN classification in the learned space for held-out time (left) and held-out channels (center), plus a 2D PCA view of test-segment embeddings (right). **B) MSC:** The same evaluations for MSC. Embeddings are organized on a hypersphere with broader, angle-separated region clouds. Dotted ellipses highlight region-consistent structure in both methods.

nels using their coordinate system. An ablation study was done on the type of the coordinate system used, namely, functional embeddings or MNI coordinates (Fig. 6A). We compared three otherwise-identical transformers that differed only in the channel embedding (coordinate system) used at tokenization: MNI coordinates, Functional-1 (PSC), and Functional-2 (MSC). Performance was summarized by the Pearson correlation ($r$) between predicted and true waveforms, computed per target channel and window, then aggregated per subject using Fisher-z averaging. Importantly, across subjects, functional embeddings significantly improved reconstruction accuracy relative to MNI (purely anatomical) for both hemispheres (Fig. 6B). Per-subject dot plots (Fig. 6B) show consistent upward shifts from MNI to Functional-1 and Functional-2. Over all subjects (box plots fig. 6B), Functional-2 outperformed MNI with statistical significance (paired test on subject-level $r$; $p \approx 0.002$ in this cohort), while Functional-1 showed a positive trend that was smaller and not reliably significant. Importantly, all results were obtained without subject-ID tokens, subject-specific heads, or fine-tuning, i.e., a single shared model across all subjects. Qualitatively, functional embeddings also produced more differentiated reconstructions across simultaneously recorded VO channels (Fig. 6B). Because the four VO electrodes often shared nearly identical or uncertain MNI coordinates, the MNI-based model tended to generate similar waveforms for targets. In contrast, Functional-1/2 leveraged each channel's functional identity, yielding channel-specific predictions that better matched the heterogeneous ground-truth VO activity. We benchmarked our model against subject-specific baselines trained separately for each individual, including a linear causal multi-output FIR, a Temporal Convolutional Network (TCN), a 2-layer GRU, a CopyBest correlation baseline, and a Zero predictor (details in Appendix A.6). Across both left and right hemispheres, the **Transformer + Functional Embedding-2 (FUNC2)** trained on pooled multi-subject data achieved the highest reconstruction accuracy, statistically significantly outperforming all per-subject baselines (paired two-sided $t$-tests with Holm and BH/FDR correction; all corrected $p < 0.001$). These results indicate that, under a unified functional coordinate system, the proposed transformer effectively aggregates information across subjects to learn stronger inter-areal relationships for signal reconstruction.

### 4.4.1 ADDITIONAL ANALYSIS

Additional results reported in Appendix A.7 indicate that excluding individual subjects from functional-embedding training does not impair downstream task performance (masked-region reconstruction) for those subjects.

To further explore cross-subject data aggregation in the downstream task, we performed a scalability analysis in which we gradually increased the number of subjects used to train the Functional Transformer. We observed strong initial gains between 5–8 subjects, with continued improvements up to 20 subjects (Appendix A.10).

Furthermore, an additional control analysis (Appendix A.9) shows that providing only anatomical region identity is insufficient to reproduce the performance gains achieved via functional embeddings. The functional encoder therefore learns structure that is not reducible to region labels, supporting our claim that the functional representation captures data-driven and meaningful functional organization that improves downstream masked-region reconstruction over purely anatomical coordinates (MNI or region labels).

Finally, to demonstrate the utility of this coordinate system in a more clinically relevant setting, we performed an additional decoding task: *predicting per-window spectral band power*. Results reported in Appendix A.8 show that functional coordinates consistently yield substantially higher correlation between predicted and true band power compared to a transformer using anatomical coordinates.

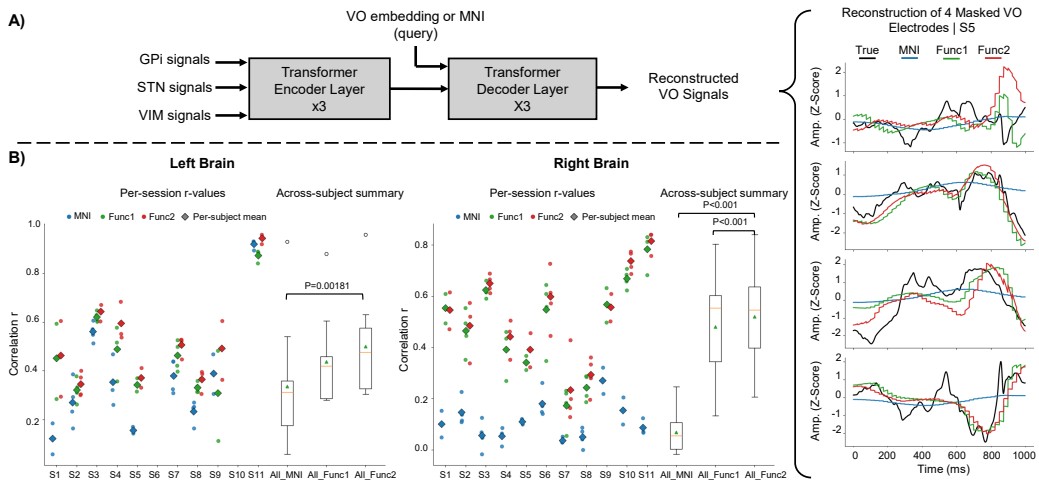

Figure 6: *Ablation of coordinate systems used in masked–region reconstruction across subjects.* **(A)** Schematic: GPi/STN/VIM signals enter a shared encoder as source tokens; VO channels are queried via either MNI or functional embeddings, and the decoder reconstructs VO waveforms. Example of 4 synchronous VO channels reconstruction attempts with 3 different coordinate systems are provided on the right side. **(B)** Per–subject results (S1–S11): per–session points and subject means show consistent gains for Functional–1/2 over MNI for both brain sides. Across–subject box plots (All MNI / All Func1 / All Func2) summarize higher correlation $r$ for functional embeddings; Functional–2 is significantly better than MNI. All models use a single shared trunk with no subject IDs, subject–specific heads, or fine–tuning.

## 5 DISCUSSION

We asked whether neural signals can be aligned across subjects by *function* rather than standard coordinates, and whether such a representation can support scalable transformer models operating over variable and sparse electrode sets. Our findings support both claims. A contrastively trained encoder learns a subject-agnostic *functional identity* for each channel: short segments from the same deep-brain region cluster in a locality-sensitive latent space and generalize across time, channels, and subjects. Conditioning transformers on these functional tokens enables cross-subject aggregation without subject IDs or subject-specific heads, yielding improved masked-region reconstruction over atlas-based (MNI) tokenization. Together, this suggests that data-derived coordinates provide a more reliable backbone for heterogeneous intracranial datasets than static anatomical alignment.

Two recent directions highlight this distinction. Mentzelopoulos et al. (2024) mix electrode signals with MNI coordinates and finish with subject-specific heads. Our approach removes both subject

identifiers and atlas coordinates, replacing them with a learned functional identity. This matters when localization is uncertain or overlapping; functional tokens consistently yield clearer reconstructions than MNI tokens under otherwise identical transformers, consistent with their own finding that positional encodings did not improve performance. Chau et al. (2025) propose Population Transformer (PopT), which aggregates frozen single-channel embeddings with positional encodings. Our pipeline is complementary: we pretrain the *channel identity* itself via contrastive learning, producing region-consistent functional coordinates that can serve as tokens for any aggregator. Conceptually, PopT learns the population-level *aggregator*, while our method learns the population-level *coordinate system*.

A functional coordinate system enables (i) subject-agnostic pooling across irregular montages, (ii) plug-and-play tokenization for large transformer backbones, and (iii) reusable pretraining across tasks such as decoding, forecasting, and imputation. Because our recordings were collected under flexible rest/movement recording sessions rather than narrowly defined tasks, the approach extends naturally to larger, more varied datasets—the setting where transformers excel.

Future work should relax reliance on region labels during contrastive training, exploring self- and weakly supervised objectives that preserve functional locality at scale. Extending beyond basal ganglia-thalamic circuits to cortical ECoG and spikes will test generality. Finally, a fuller head-to-head comparison against population-level pretraining frameworks such as PopT remains important. Bridging the two directions—functional coordinates for channel identity and ensemble-level pretraining—may yield the best of both worlds.

## ETHICS STATEMENT

All human neural data analyzed in this study were collected under protocols approved by the Institutional Review Boards (IRBs) of the participating medical centers. Written informed consent was obtained from all participants or their legal guardians (for minors), with assent obtained from children when appropriate. Data handling complied with HIPAA and institutional privacy policies: recordings were de-identified prior to analysis, stored on secure, access-controlled systems, and used solely for research purposes. This work does not involve clinical decision-making, patient intervention beyond standard clinical care, or vulnerable-population recruitment beyond the clinical cohort already undergoing DBS evaluation. No attempt was made to re-identify participants. Due to ethical and regulatory constraints governing research with human subjects, the raw neural recordings cannot be publicly released; de-identified data may be shared upon reasonable request and execution of a data use agreement consistent with IRB approvals. The authors affirm compliance with the ICLR Code of Ethics and disclose no conflicts of interest or external sponsorship that could influence the work.

## REPRODUCIBILITY STATEMENT

All code used in this study is released to support reproducibility at `https://github.com/ICLR-Functional-Embedding/ICLR2026_Functional_Map`. Complete and detailed explanation of datasets used, data preprocessing, training details and training resources and are provided in Appendix.

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

Table A.1: *Demographic characteristics:* 20 male subjects, between 5 and 24 years old. H-ABC: Hypomyelination with Atrophy of the Basal Ganglia and Cerebellum; CP: Cerebral palsy; GA1: Glutaric aciduria type 1; MEPAN: Mitochondrial Enoyl CoA Reductase Protein-Associated Neurodegeneration; KMT2B: K-specific Methyltransferase 2B; PKAN: Pantothenate Kinase-Associated Neurodegeneration; MYH2: Myosin Heavy Chain 2; GPi: globus pallidus internus; VoaVop: ventral oralis anterior-posterior; STN: subthalamic nucleus; VA: ventral anterior; VPL: ventral posterolateral; VIM: ventral intermediate; PPN: Pedunculopontine nucleus; SNR: Substantia nigra pars reticulata; CM: Centromedian nucleus; CL: Centrolateral nucleus.

| Subject | Etiology | Implanted Regions |
|---------|----------|-------------------|
| 1 | Tourette | NA, VoaVop, VIM, STN, CMP, GPi |
| 2 | CP | VA, VoaVop, VIM, STN, GPi |
| 3 | GA1 | VA, VoaVop, VIM, STN, GPi |
| 4 | MEPAN | VA, VoaVop, VIM, PPN, STN, GPi |
| 5 | MEPAN | VA, VoaVop, VIM, PPN, STN, GPi |
| 6 | Unknown | VA, VoaVop, PPN, STN, GPi |
| 7 | KMT2B | VA, VoaVop, VIM, PPN, STN, GPi |
| 8 | CP | VA, VoaVop, VIM, PPN, STN, GPi |
| 9 | GA1 | VA, VoaVop, VIM, PPN, STN, GPi |
| 10 | PKAN | VoaVop, VIM, PPN, SNR, STN, GPi |
| 11 | CP | VA, VoaVop, VIM, PPN, SNR, STN, GPi |
| 12 | MYH2 | VoaVop, VIM, PPN, SNR, STN, GPi |
| 13 | CP | VoaVop, VIM, PPN, SNR, STN, GPi |
| 14 | GA1 | VoaVop, VIM, PPN, SNR, STN, GPi |
| 15 | CP | VoaVop, VIM, STN, GPi |
| 16 | CP | VA, VoaVop, VIM, PPN, STN, GPi |
| 17 | KMT2B | VoaVop, VIM, CM, STN, GPi |
| 18 | CP | VoaVop, VIM, CM, STN, GPi |
| 19 | CP | VoaVop, VIM, CM, SNR, STN, GPi |
| 20 | KMT2B | VoaVop, VIM, CM, CL, STN, GPi |

Xiaolong Wu, Guangye Li, Xin Gao, Benjamin Metcalfe, and Dingguo Zhang. Channel selection for stereo-electroencephalography (seeg)-based invasive brain-computer interfaces using deep learning methods. *IEEE Transactions on Neural Systems and Rehabilitation Engineering*, 32:800–811, 2024.

Joel Ye and Chethan Pandarinath. Representation learning for neural population activity with neural data transformers. *arXiv preprint arXiv:2108.01210*, 2021.

Joel Ye, Jennifer Collinger, Leila Wehbe, and Robert Gaunt. Neural data transformer 2: multi-context pretraining for neural spiking activity. biorxiv, 2023.

# A APPENDIX

## A.1 DATASET AND CLINICAL METADATA

**Subjects.** Twenty (20) subjects who underwent DBS surgery for the treatment of dystonia participated in this study (Table A.1). The diagnosis of dystonia was established by a pediatric movement disorder specialist (Anonymous) using standard criteria (Anonymous, 2003). All patients provided signed informed consent for surgical procedures in accordance with standard hospital practice at participating hospitals (anonymized for review). The study protocol was approved by the institutional review boards (IRBs) of both hospitals. The patients, or parents of minor patients, also signed an informed consent to the research use of electrophysiological data and provided Health Insurance Portability and Accountability Act (HIPAA) authorization for the research use of protected health information.

Table A.2: *Recording coverage by subject and brain region:* "Recording time" sums recording lengths over all sessions for each subject. "Electrode hours" multiply each subject's total recording time by that subject's number of recorded channels across regions.

| Subject (sessions) | GPi | PPN | SNR | STN | VA | VIM | VO | Recording time (hrs) | Electrode hours |
|---|---|---|---|---|---|---|---|---|---|
| S1 (2 sessions) | 19 | 0 | 0 | 6 | 0 | 9 | 6 | 1.60 | 63.81 |
| S2 (3 sessions) | 12 | 0 | 0 | 6 | 6 | 6 | 6 | 0.36 | 12.97 |
| S3 (3 sessions) | 12 | 0 | 0 | 6 | 6 | 6 | 6 | 0.37 | 13.40 |
| S4 (2 sessions) | 18 | 12 | 0 | 6 | 8 | 0 | 8 | 0.23 | 12.06 |
| S5 (4 sessions) | 18 | 6 | 0 | 6 | 8 | 0 | 8 | 0.62 | 28.36 |
| S6 (4 sessions) | 8 | 3 | 0 | 6 | 8 | 0 | 6 | 0.47 | 14.46 |
| S7 (3 sessions) | 12 | 6 | 0 | 6 | 12 | 0 | 6 | 0.53 | 22.29 |
| S8 (4 sessions) | 12 | 6 | 0 | 6 | 6 | 6 | 6 | 0.46 | 19.42 |
| S9 (2 sessions) | 21 | 6 | 0 | 8 | 11 | 0 | 4 | 0.24 | 11.94 |
| S10 (4 sessions) | 29 | 9 | 0 | 10 | 0 | 0 | 4 | 0.44 | 23.07 |
| S11 (5 sessions) | 14 | 9 | 14 | 6 | 15 | 0 | 8 | 0.53 | 35.24 |
| S12 (3 sessions) | 25 | 9 | 0 | 10 | 0 | 0 | 4 | 0.30 | 14.64 |
| S13 (2 sessions) | 8 | 9 | 12 | 4 | 0 | 0 | 4 | 0.55 | 20.27 |
| S14 (2 sessions) | 15 | 8 | 6 | 6 | 6 | 0 | 6 | 0.25 | 11.74 |
| S15 (4 sessions) | 19 | 0 | 0 | 8 | 0 | 9 | 2 | 0.61 | 23.27 |
| S16 (4 sessions) | 15 | 10 | 0 | 6 | 8 | 0 | 8 | 0.63 | 29.51 |
| S17 (4 sessions) | 18 | 0 | 0 | 10 | 0 | 8 | 4 | 0.50 | 20.18 |
| S18 (2 sessions) | 20 | 0 | 0 | 11 | 0 | 10 | 6 | 0.25 | 11.84 |
| S19 (4 sessions) | 26 | 12 | 9 | 9 | 0 | 0 | 2 | 0.51 | 29.80 |
| S20 (3 sessions) | 20 | 0 | 0 | 12 | 0 | 10 | 8 | 0.49 | 24.62 |
| **Total** | **341** | **105** | **41** | **148** | **94** | **64** | **112** | **9.96** | **442.86** |

**Experimental protocol and recording sessions.** Participants performed self-paced finger-to-nose reaching or hand-squeeze movements, depending on motor ability. The clinical team administered short blocks consisting of ~60 s of active movement followed by ~30 s of rest, repeated at least four times per hand. Trajectory and speed were unconstrained, and block durations were adjusted as needed to accommodate each participant. For all analyses, recordings were not segmented by trials or active/rest windows; instead, we used the entire continuous LFP recordings from each session. This yields behaviorally diverse, non-stationary signals, underscoring the advantage of our representation-learning approach, which remains robust to noise and variability in naturalistic clinical settings.

**Recording electrodes and spatial coverage.** Local field potentials (LFPs) were recorded from high-impedance microwire ("micro-contact") electrodes on temporary Stereo-EEG (SEEG) depth leads (AdTech MM16C; AdTech Medical Instrument Corp., Oak Creek, WI, USA) implanted with standard stereotactic procedures Anonymous (2024b; 2019). Up to 12 SEEG leads were placed per subject to sample candidate DBS targets. Each lead includes six low-impedance ring macro-contacts (1–2 kΩ; 2 mm height) and ten high-impedance 50-$\mu$m micro-contacts (70–90 kΩ). Micro-contact signals were sampled at 24,414 Hz using a PZ5M 256-channel digitizer and RZ2 processor, with data stored on an RS4 system (Tucker-Davis Technologies, Alachua, FL, USA). All analyses in this paper use LFPs from the micro-contacts; macro-contact recordings were not used.

Table A.2 summarizes electrode spatial coverage per subject (S1–S20), listing the number of unique recording channels in each brain region alongside the total recording time and electrode-hours. Brain-region assignments were performed by an expert via manual review of co-registered MRI–CT volumes. Recording time is computed as the sum of session lengths for each subject, while electrode-hours equal each subject's total recording time multiplied by their total channel count. Across all subjects, the dataset comprises 9.96 recording hours and 442.86 electrode-hours.

**MNI coordinate extraction.** MNI coordinates for a total of 11 subjects yielding approximately 600 macro-contact were extracted. Structural T1-weighted (T1W) MRI scans for all 11 subjects were nonlinearly normalized to MNI space using the Advanced Normalization Tools (ANTs) framework (Avants et al., 2008), a widely adopted tool for high-accuracy nonlinear image registration

(Tustison et al., 2019; Bartel et al., 2019). Postoperative CT scans were rigidly co-registered to each subject's preoperative T1W MRI using the FLIRT (FMRIB's Linear Image Registration Tool) module of the FMRIB Software Library (FSL). The nonlinear warping transformations obtained from ANTs normalization of the T1W images were subsequently applied to the co-registered CT scans, thereby aligning them to MNI space. Following spatial normalization, approximately 600 macro-contacts were localized across all patients. Contact identification was performed using histogram analysis of CT metal artifacts, with the voxel exhibiting the maximum CT intensity within each artifact region selected as the contact center coordinate. These standardized MNI coordinates of macro-contacts were then used to generate known relative positions for micro-contacts which recorded the LFP signals. micro-contact locations were then used as inputs for the machine learning models. Figures A.1 shows all the micro-electrode locations from 11 subjects relative to 4 deep brain regions.

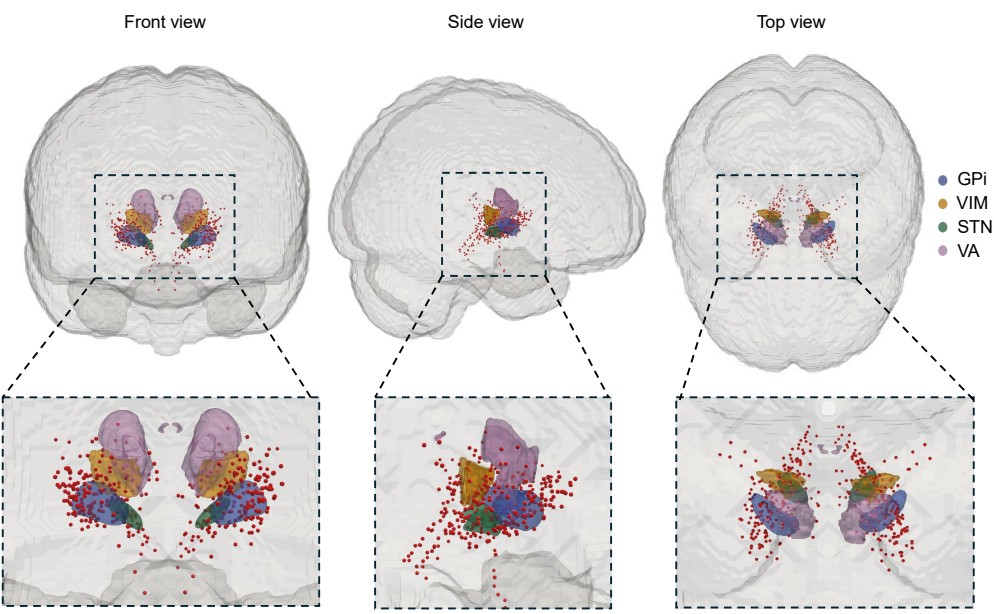

Figure A.1: *3 views of aggregated micro-contacts locations over 11 subjects relative to 4 deep brain regions.*

## A.2 ADDITIONAL SIGNAL PREPROCESSING DETAILS

All preprocessing was performed per channel with zero-phase filtering. We describe the common steps followed by branch-specific steps for the functional encoder and transformer experiments.

**Common steps.**

1. **Anti-aliasing and resampling.** A 4th-order Butterworth low-pass filter with cutoff 500 Hz was applied, then signals were resampled to 1,000 Hz .

2. **Line-noise removal.** Narrowband notches were applied at 60, 120, and 180 Hz using an IIR notch filter with quality factor $Q=30$.

3. **Session-wise normalization.** For each session, the signal was standardized by subtracting the session mean and dividing by the session standard deviation.

**Functional encoder (contrastive) pipeline.**

1. **Windowing.** Continuous recordings were segmented into non-overlapping 10 s windows.

2. **Robust artifact rejection.** Let $x$ denote the full-session signal values for a channel. Define

$$\text{med} = \text{median}(x), \quad \widehat{\sigma} = 1.4826 \cdot \text{median}(|x - \text{med}|) \quad \text{(MAD-based robust std)}.$$

Windows were discarded if any sample fell outside $[\text{med} - \tau\widehat{\sigma}, \ \text{med} + \tau\widehat{\sigma}]$, with threshold $\tau = 10$ in our implementation.

3. **Splits.** Non-overlapping 10 s windows were assigned to splits without shuffling in chronological order: the first 70% of windows to training, the next 15% to validation, and the final 15% to test. Entire windows were assigned to a single split (no window was divided across sets).

**Transformer pipeline.**

1. **Low-pass filtering.** After the notch filters, a 2nd-order Butterworth low-pass filter with 50 Hz cutoff was applied.

2. **Chronological splits.** The first 70% of each session (in time) was used to construct training examples; the next 15% and final 15% were used for validation and test, respectively.

## A.3 ADDITIONAL DETAILS ON SIMULATION STUDY

To evaluate our embedding methods in a fully controlled setting, we constructed a synthetic dataset designed to reflect known region-specific neural signatures while allowing precise control over subject-, session-, and electrode-level variability.

The dataset includes:

- $N = 10$ subjects
- $S = 2$ sessions per subject
- $R = 5$ brain regions per session: GPi, STN, VO, Motor Cortex, and VIM
- $C = 4$ electrodes (channels) per region
- $f_s = 1000$ Hz sampling rate

Each recording channel was synthesized by combining a region-specific neural signal generator with additive Gaussian noise, modulated by three hierarchical levels of parameters: subject-level, session-level, and electrode-level.

**1. Region-Specific Neural Signatures**

Each brain region was associated with one or more characteristic oscillatory components:

Each oscillatory component was implemented using sine waves modulated by windowed envelopes (e.g., Hann windows for bursts), with frequency jitter and amplitude variation across subjects and channels.

| Region | Signature Components |
|--------|---------------------|
| GPi | Beta bursts ($\sim$20 Hz) |
| STN | Beta bursts + HFOs ($\sim$150 Hz) |
| VO | Spindle bursts ($\sim$12–16 Hz) |
| Motor | Continuous gamma activity ($\sim$40 Hz) |
| VIM | Slow oscillations ($\sim$1 Hz) |

## 2. Parameter Hierarchy

We define the following hierarchical parameter sets:

**Subject-Level Parameters: $\theta_{\text{subject}}$** Sampled once per subject, controlling global oscillation strength and structure:

$$\theta_{\text{subject}} = \{\beta_{\text{gain}}, \beta_{\text{freq\_offset}}, \beta_{\text{burst\_prob}}, \text{spindle}_{\text{gain}}, \text{spindle}_{\text{freq\_offset}}, \text{spindle}_{\text{burst\_prob}}, \gamma_{\text{gain}}, \gamma_{\text{freq\_offset}},$$
$$\text{HFO}_{\text{gain}}, \text{HFO}_{\text{freq\_offset}}, \text{slow}_{\text{gain}}, \text{slow}_{\text{freq\_offset}}, \text{noise}_{\text{level}}\}$$

Sampled from the following distributions:

$$\beta_{\text{gain}} \sim \mathcal{U}(0.8,\ 1.5) \qquad \beta_{\text{freq\_offset}} \sim \mathcal{U}(-2.0,\ 2.0) \qquad \beta_{\text{burst\_prob}} \sim \mathcal{U}(0.3,\ 0.7)$$
$$\text{spindle}_{\text{gain}} \sim \mathcal{U}(0.8,\ 1.3) \quad \text{spindle}_{\text{freq\_offset}} \sim \mathcal{U}(-1.0,\ 1.0) \quad \text{spindle}_{\text{burst\_prob}} \sim \mathcal{U}(0.2,\ 0.5)$$
$$\gamma_{\text{gain}} \sim \mathcal{U}(0.3,\ 0.7) \qquad \gamma_{\text{freq\_offset}} \sim \mathcal{U}(-5.0,\ 5.0)$$
$$\text{HFO}_{\text{gain}} \sim \mathcal{U}(0.1,\ 0.3) \qquad \text{HFO}_{\text{freq\_offset}} \sim \mathcal{U}(-10.0,\ 10.0)$$
$$\text{slow}_{\text{gain}} \sim \mathcal{U}(0.8,\ 1.2) \qquad \text{slow}_{\text{freq\_offset}} \sim \mathcal{U}(-0.3,\ 0.3)$$
$$\text{noise}_{\text{level}} \sim \mathcal{U}(0.3,\ 0.4)$$

**Session-Level Parameters: $\theta_{\text{session}}$** Sampled per session:

$$\theta_{\text{session}} = \{\text{session\_noise} \sim \mathcal{U}(0.05,\ 0.15)\}$$

**Electrode-Level Parameters: $\theta_{\text{electrode}}$** Sampled per electrode:

$$\theta_{\text{electrode}} = \{\text{gain} \sim \mathcal{U}(0.9,\ 1.1),\ \text{electrode\_noise} \sim \mathcal{U}(0.03,\ 0.1)\}$$

## 3. Signal Model

For a given region $r$, session $s$, and electrode $e$, the generated signal is defined as:

$$\text{Signal}_{r,s,e}(t) = \text{Signature}_r(t;\ \theta_{\text{subject}}, \theta_{\text{electrode}}) + \mathcal{N}\left(0,\ \sigma_{\text{subject}}^2 + \sigma_{\text{session}}^2 + \sigma_{\text{electrode}}^2\right)$$

Each component of the signal is constructed from filtered oscillatory patterns with gain-scaled amplitudes, frequency offsets, and additive Gaussian noise aggregated from all three levels. The pipeline for generating simulated dataset is depicted in fig. A.2.

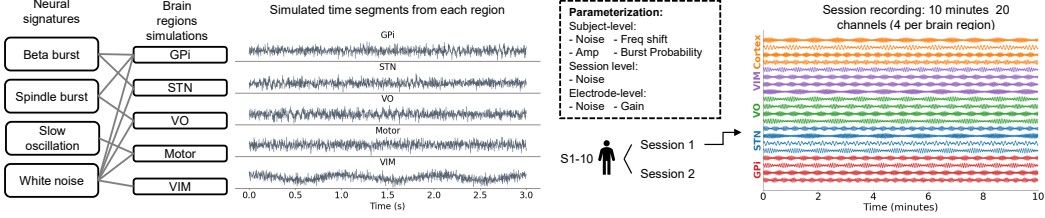

Figure A.2: *Simulation dataset.* pipeline: each region is assigned a parameterized neural signature; recordings are generated across subjects/sessions with gain, frequency shifts, and noise parameters.

## A.4 CNN FUNCTIONAL ENCODER: ARCHITECTURE AND TRAINING

**Setup.** We learn a subject-agnostic 32D embedding of 1D neural time series using convolutional neural network.

**Single-session encoder.** The encoder $f_\theta$ is a 1D CNN with following blocks:

$$\text{Conv1d}(1, 64, k{=}15, s{=}4) \rightarrow \text{BN}(64) \rightarrow \text{GELU} \rightarrow \text{MaxPool}(3, s{=}2) \rightarrow \text{Dropout}$$
$$\text{Conv1d}(64, 64, k{=}11, s{=}2) \rightarrow \text{BN}(64) \rightarrow \text{GELU} \rightarrow \text{MaxPool}(3, s{=}2) \rightarrow \text{Dropout}$$
$$\text{Conv1d}(64, 64, k{=}7, s{=}2) \rightarrow \text{BN}(64) \rightarrow \text{GELU} \rightarrow \text{MaxPool}(3, s{=}2) \rightarrow \text{Dropout}$$
$$\text{Conv1d}(64, 64, k{=}5, s{=}2) \rightarrow \text{BN}(64) \rightarrow \text{GELU} \rightarrow \text{MaxPool}(3, s{=}2) \rightarrow \text{Dropout}$$
$$\text{AdaptiveAvgPool1d}(10) \;\rightarrow\; \text{Flatten}\,(10{\times}64) \;\rightarrow\; \text{MLP: } \mathbb{R}^{640} \rightarrow \mathbb{R}^{32} \rightarrow \mathbb{R}^{32}.$$

The output embedding is $\ell_2$-normalized. This configuration has 117,504 trainable parameters.

**Training.** We train for 100 epochs using AdamW (Loshchilov & Hutter, 2019) (weight decay $10^{-4}$, $\beta_1{=}0.9$, $\beta_2{=}0.99$) with initial learning rate lr=0.01, batch size 128, and a ReduceLROnPlateau scheduler (mode=min, patience=5, factor=0.5, $\text{lr}_{\min}{=}10^{-5}$). Dropout in conv blocks is 0.5. The embedding head applies a small dropout before the final linear layer and outputs unit-norm vectors.

**Multi-subject / multi-session encoder** For pooled training across subjects and sessions we use a deeper/wider encoder with the same design principles, increasing channels and depth before the global pooling step:

$$\text{Conv1d}(1, 64, k{=}15, s{=}4) \rightarrow \text{BN}(64) \rightarrow \text{GELU} \rightarrow \text{MaxPool}(3, s{=}2) \rightarrow \text{Dropout}$$
$$\text{Conv1d}(64, 128, k{=}11, s{=}2) \rightarrow \text{BN}(128) \rightarrow \text{GELU} \rightarrow \text{MaxPool}(3, s{=}2) \rightarrow \text{Dropout}$$
$$\text{Conv1d}(128, 128, k{=}11, s{=}2) \rightarrow \text{BN}(128) \rightarrow \text{GELU} \rightarrow \text{MaxPool}(3, s{=}2) \rightarrow \text{Dropout}$$
$$\text{Conv1d}(128, 128, k{=}11, s{=}2) \rightarrow \text{BN}(128) \rightarrow \text{GELU} \rightarrow \text{MaxPool}(3, s{=}2) \rightarrow \text{Dropout}$$
$$\text{Conv1d}(128, 128, k{=}7, s{=}2) \rightarrow \text{BN}(128) \rightarrow \text{GELU} \rightarrow \text{Dropout}$$
$$\text{Conv1d}(128, 128, k{=}5, s{=}2) \rightarrow \text{BN}(128) \rightarrow \text{GELU} \rightarrow \text{Dropout}$$
$$\text{AdaptiveAvgPool1d}(10) \;\rightarrow\; \text{Flatten}\,(10{\times}128) \;\rightarrow\; \text{MLP: } \mathbb{R}^{1280} \rightarrow \mathbb{R}^{32} \rightarrow \mathbb{R}^{32}.$$

The output embedding remains $F{=}32$ and is $\ell_2$-normalized. This configuration has 694,464 trainable parameters. Dropout in conv blocks is set to 0.3.

**Training.** We use the same optimizer family and regularization as above (AdamW with weight decay $10^{-4}$, $\beta_1{=}0.9$, $\beta_2{=}0.99$), increasing the batch size to 1024 to leverage cross-subject aggregation. Learning-rate scheduling follows the same ReduceLROnPlateau policy; the initial learning rate matches the single-session setup unless otherwise stated. The embedding head outputs unit-norm vectors used by the contrastive criteria in the main text.

## A.5 DETAILS OF THE FUNCTIONAL TRANSFORMER

The functional embedding of each electrode serves as a subject-agnostic channel identity. For every channel, the first 10 s snippet (from training dataset) is fed to the embedding encoder to produce its functional vector; this vector is then projected to the model width and reused unchanged for all tokens originating from that channel (across time patches and windows). We fuse the projected embedding with time–patch features and a time-only positional code to form the tokens consumed by the transformer. Variable channel counts are handled with padding masks.

**Masked-Region Reconstruction Setup.** For each window we have multichannel LFP signals $\mathbf{X} \in \mathbb{R}^{C \times T}$ and a per–channel *functional embedding* matrix $\mathbf{E} \in \mathbb{R}^{C \times F}$ with $F{=}32$. At evaluation, a region $\mathcal{R}$ (e.g., VO) is fully masked: its channels provide no signal. Let $\mathcal{S} = \{c : c \notin \mathcal{R}\}$ be *sources* and $\mathcal{T} = \{k : k \in \mathcal{R}\}$ be *targets*. Given source signals $\{\mathbf{x}_c\}_{c \in \mathcal{S}}$ and source and target embeddings $\{\mathbf{e}_c\}_{c \in \mathcal{S} \cup \mathcal{T}}$, the model reconstructs target signals $\{\widehat{\mathbf{y}}_k\}_{k \in \mathcal{T}}$.

**Functional tokens.** For each channel, a single 10 s snippet (from training data) is fed to the embedding encoder to produce its functional vector $\mathbf{e}_c \in \mathbb{R}^{32}$. This vector is linearly mapped to model width $\widehat{\mathbf{f}}_c \in \mathbb{R}^d$ and then reused unchanged for every token originating from that channel (across patches and windows). Layer normalization and learned type embeddings follow standard practice and are omitted from equations for clarity.

**Tokenization of source signals.** A 1D conv "tokenizer" converts each source $\mathbf{x}_c$ into $P$ time patches. Let $t \in \{1, \ldots, P\}$ index the patch. Formally,

$$\mathbf{Z}_c = \text{Conv1d}(\mathbf{x}_c; L_{\text{patch}}, S) \in \mathbb{R}^{P \times d}, \qquad P \approx \left\lfloor \frac{T - L_{\text{patch}}}{S} \right\rfloor + 1,$$

where $d$ is both the number of conv filters and the Transformer width. $\mathbf{Z}_{ct} \in \mathbb{R}^d$ is the $d$-dim temporal feature for channel $c$ at patch $t$. We *broadcast* the functional identity $\mathbf{f}_c \in \mathbb{R}^d$ to all patches of channel $c$ and *concatenate* it with the temporal features, $[\mathbf{Z}_{ct} \| \mathbf{f}_c] \in \mathbb{R}^{2d}$. A dense layer *reduces back to width $d$* and we add a *time-only* positional code $\mathbf{P}_t^{(t)} \in \mathbb{R}^d$, yielding a source token of shape:

$$\widetilde{\mathbf{Z}}_{ct} \in \mathbb{R}^d \qquad (c \in \mathcal{S}, \ t = 1, \ldots, P).$$

Stacking tokens across channels and patches produces the encoder input sequence

$$\mathbf{X}_{\text{src}} = [\widetilde{\mathbf{Z}}_{ct}]_{c \in \mathcal{S}, \ t=1..P} \in \mathbb{R}^{S_{\text{tok}} \times d}, \quad S_{\text{tok}} = |\mathcal{S}| \, P,$$

with a key-padding mask $\mathbf{m}_{\text{src}} \in \{0, 1\}^{S_{\text{tok}}}$ to ignore padded (absent) sources within a batch (due to variable number of channels present in the dataset).

**Tokenization of query signals (masked reconstruction targets).** Targets have no signal, so temporal features are replaced with a *learned query base* $\mathbf{U}_t \in \mathbb{R}^d$ (one $d$-vector per patch index $t$). For each target channel $k \in \mathcal{T}$ and patch $t$, we concatenate the base with the functional embedding of that channel $[\mathbf{U}_t \| \mathbf{f}_k] \in \mathbb{R}^{2d}$, reduce back to width $d$, and add the same time-only positional code, producing query tokens of shape:

$$\mathbf{Q}_{kt} \in \mathbb{R}^d \qquad (k \in \mathcal{T}, \ t = 1, \ldots, P).$$

Stacking all targets and patches forms the decoder input sequence

$$\mathbf{Q} = [\mathbf{Q}_{kt}]_{k \in \mathcal{T}, \ t=1..P} \in \mathbb{R}^{T_{\text{tok}} \times d}, \quad T_{\text{tok}} = |\mathcal{T}| \, P,$$

with a query-padding mask $\mathbf{m}_{\text{tgt}} \in \{0, 1\}^{T_{\text{tok}}}$ to ignore padded (absent) targets.

**Encoder-decoder attention.** We use pre-LN Transformer blocks with $h$ heads and per–head size $d_h = d/h$. In particular multi-head self-attention (MHSA) and multi-head attention (MHA) blocks are used where :

$$\text{MHSA}(\mathbf{X}) = \text{Concat}_h \big[ \text{Attn}(\mathbf{Q}_i, \mathbf{K}_i, \mathbf{V}_i) \big] \mathbf{W}^O, \quad \mathbf{Q}_i = \mathbf{X}\mathbf{W}_i^Q, \ \mathbf{K}_i = \mathbf{X}\mathbf{W}_i^K, \ \mathbf{V}_i = \mathbf{X}\mathbf{W}_i^V,$$

where $\mathbf{W}^O$ projects $h$ concatenated outputs to model width ($d$), $i$ denotes a specific head from $h$ heads, and Attn defined as:

$$\text{Attn}(\mathbf{Q}_i, \mathbf{K}_i, \mathbf{V}_i) = \text{softmax}\left( \frac{\mathbf{Q}_i \mathbf{K}_i^\top}{\sqrt{d_h}} \right) \mathbf{V}_i.$$

MHA follows identical formulation where query ($\mathbf{Q}_i$) comes from a source other than $\mathbf{X}$.

**Encoder (self-attn on sources).**
With source tokens $\mathbf{X}_{\text{src}} \in \mathbb{R}^{S_{\text{tok}} \times d}$,

$$\underbrace{\mathbf{Q}^{\text{enc}} = \mathbf{X}_{\text{src}} \mathbf{W}_{\text{enc}}^Q, \quad \mathbf{K}^{\text{enc}} = \mathbf{X}_{\text{src}} \mathbf{W}_{\text{enc}}^K, \quad \mathbf{V}^{\text{enc}} = \mathbf{X}_{\text{src}} \mathbf{W}_{\text{enc}}^V}_{\in \mathbb{R}^{S_{\text{tok}} \times d}} \Rightarrow \mathbf{M} = \text{MHSA}(\mathbf{X}_{\text{src}}) \in \mathbb{R}^{S_{\text{tok}} \times d},$$

with key-padding mask $\mathbf{m}_{\text{src}}$ applied inside the softmax.

**Decoder (self-attn on queries, then cross-attn to memory).**
First, self-attention over query tokens $\mathbf{Q} \in \mathbb{R}^{T_{\text{tok}} \times d}$:

$$\underbrace{\mathbf{Q}^{\text{dec}} = \mathbf{Q}\,\mathbf{W}_{\text{dec}}^Q, \ \mathbf{K}^{\text{dec}} = \mathbf{Q}\,\mathbf{W}_{\text{dec}}^K, \ \mathbf{V}^{\text{dec}} = \mathbf{Q}\,\mathbf{W}_{\text{dec}}^V}_{\in \mathbb{R}^{T_{\text{tok}} \times d}} \Rightarrow \widetilde{\mathbf{Q}} = \text{MHSA}(\mathbf{Q}) \in \mathbb{R}^{T_{\text{tok}} \times d},$$

masking padded query tokens via $\mathbf{m}_{\text{tgt}}$. Next, *cross-attention* where queries attend to encoder memory:

$$\underbrace{\mathbf{Q}^{\text{qry}} = \widetilde{\mathbf{Q}}\,\mathbf{W}_{\text{qry}}^Q}_{\in \mathbb{R}^{T_{\text{tok}} \times d}}, \quad \underbrace{\mathbf{K}^{\text{mem}} = \mathbf{M}\,\mathbf{W}_{\text{mem}}^K, \ \mathbf{V}^{\text{mem}} = \mathbf{M}\,\mathbf{W}_{\text{mem}}^V}_{\in \mathbb{R}^{S_{\text{tok}} \times d}} \Rightarrow \mathbf{H} = \text{MHA}(\widetilde{\mathbf{Q}}, \mathbf{M}, \mathbf{M}) \in \mathbb{R}^{T_{\text{tok}} \times d},$$

with key-padding mask $\mathbf{m}_{\text{src}}$ applied to the memory keys.

Stacking $L_{\text{enc}}$ and $L_{\text{dec}}$ layers yields the final decoder states $\mathbf{H}$.

**Waveform head and outputs.** A linear head maps each decoder token to a patch of $L_{\text{patch}}$ samples, then patches are reshaped and concatenated along time:

$$\mathbf{H} \in \mathbb{R}^{T_{\text{tok}} \times d} \xrightarrow{\text{linear}} \mathbf{R} \in \mathbb{R}^{T_{\text{tok}} \times L_{\text{patch}}} \xrightarrow{\text{reshape}} \widehat{\mathbf{Y}} \in \mathbb{R}^{K \times P \times L_{\text{patch}}} \xrightarrow{\text{concat over } t} \widehat{\mathbf{Y}} \in \mathbb{R}^{K \times T_m}$$

$$, T_m = P\, L_{\text{patch}}$$

(An overlap–add variant with stride $S < L_{\text{patch}}$ is compatible but not used here.)

**Masking and batching.** For each window, sources exclude all channels in $\mathcal{R}$ and targets include all channels in $\mathcal{R}$. Across windows/subjects the numbers of sources and targets vary; we pad to the batch maxima and use $\mathbf{m}_{\text{src}}$ and $\mathbf{m}_{\text{tgt}}$ so attention and losses ignore padded tokens.

**Objective and optimization.** Given ground truth $\mathbf{Y} \in \mathbb{R}^{K \times T_m}$ and predictions $\widehat{\mathbf{Y}}$,

$$\mathcal{L} = \underbrace{\tfrac{1}{T_m}\|\widehat{\mathbf{Y}} - \mathbf{Y}\|_2^2}_{\text{MSE}} + \lambda\Big(1 - \rho(\widehat{\mathbf{Y}}, \mathbf{Y})\Big), \qquad \lambda = 0.05,$$

where $\rho$ is mean Pearson correlation across targets (after per-sequence standardization). The correlation term enforces *shape fidelity* (phase/temporal structure) and mitigates the tendency of pure MSE to prefer amplitude shrinkage or flat predictions under SNR or scale mismatch. No subject IDs, subject-specific heads, or fine-tuning are used; a single model is trained and evaluated across all subjects for multi-subject experiments.

**Model parameters and training.** For training on multi-session multi-subject data, we use $L_{\text{enc}} = 3$ encoder layers and $L_{\text{dec}} = 3$ decoder layers with model width $d = 128$ and $h = 4$ attention heads ($d_h = 32$). The Transformer feed-forward dimension is 384 with dropout set to 0.2. The tokenizer patch length is $L_{\text{patch}} = 25$ ms. Each model input window is 1 s of LFP, drawn from a variable number of channels (subject-dependent). Windows are extracted with a 500 ms overlap after the train/validation/test split to avoid leakage. The resulting model has approximately $1.37 \times 10^6$ trainable parameters.

Optimization uses AdamW with $\beta_1 = 0.9$, $\beta_2 = 0.99$, and $\varepsilon = 10^{-8}$. We employ per-parameter learning rates: $2 \times 10^{-4}$ for all layers except the waveform head, which uses $3 \times 10^{-4}$. Weight decay is $10^{-4}$, except it is disabled for parameters that should not be regularized (e.g., LayerNorm scales/biases, embedding and type-embedding vectors, and the learned query base), following common practice. 150 epochs and batch size of 128 was used for training.

For experiments trained on a single session (500–800 windows), we use a smaller Transformer with model width $d = 32$ and feed-forward dimension $= 32$, yielding $\sim 9.0 \times 10^4$ trainable parameters. All other architectural and optimization settings remain the same, except that we increase the learning rates by $10\times$ (i.e., $2 \times 10^{-3}$ for all layers and $3 \times 10^{-3}$ for the waveform head). Batch size and number of epochs are identical to the multi-subject setting.

### A.6 BASELINES AND ASSOCIATED TRAINING PROTOCOL

We compare our transformer against a set of standard time–series baselines that map *source* channels to *target* channels within each window. For every subject and sessions, we use identical train/validation/test splits and the same masking pipeline: all channels whose region label matches a pre-specified exclusion predicate (e.g., `VO*`) are designated as targets, while the remaining channels are sources.

**Baselines.**

- **CopyBest** (nonparametric). For each target channel, we select the single source channel with the highest training correlation (Z-scored across time) and fit a scalar gain by least squares; the prediction is a scaled copy of that source. This yields a strong correlation-based lower bound without temporal modeling.

- **FIR (linear, causal MIMO).** A multi-input multi-output finite impulse response filter that predicts targets from sources using a causal 1-D convolution with kernel size $L+1$. We set $L=64$ (thus receptive field 65 samples equivalent to 65ms), implemented as a single causal Conv1d from $C_{\text{src}}$ to $C_{\text{tgt}}$ and trained end-to-end.

- **TCN (temporal CNN).** (Bai et al., 2018) A temporal convolutional network with pointwise input/output projections and $D{=}5$ residual dilated blocks (dilations $1, 2, 4, 8, 16$), kernel size $k{=}5$, width 128, GroupNorm(1) in each block, GELU activations, and dropout 0.3. The effective receptive field approximately covers the FIR context.

- **GRU (recurrent).** A 2-layer Gated Recurrent Unit (GRU) (Cho et al., 2014) with hidden size 128 (dropout 0.3 between layers), followed by a linear projection to $C_{\mathrm{tgt}}$ at each time step.

**Optimization and early stopping.** All trainable baselines (FIR, TCN, GRU) are trained with Adam (Kingma & Ba, 2015) (learning rate $3\times10^{-4}$, weight decay $10^{-4}$), batch size 8, for up to 150 epochs with early stopping (patience 10) based on validation MSE.

**Evaluation metric.** We report Fisher-$\bar{r}$ over all valid window$\times$target pairs:

$$z_i = \operatorname{arctanh}(r_i), \quad \bar{z} = \frac{1}{N} \sum_i z_i, \quad \bar{r} = \tanh(\bar{z}),$$

where $r_i$ is the per-sample Pearson correlation (over time) between prediction and ground truth for a target channel.

**Statistics.** For across-subject comparisons we used paired two-sided $t$-tests between each baseline and our method (FUNC2), applying both Holm–Bonferroni and Benjamini–Hochberg (FDR) corrections for multiple comparisons; all corrected $p$-values were $< 0.001$, indicating statistically significant improvements.

**Baseline comparison.** Figure A.3 summarizes subject-level performance (Fisher-$\bar{r}$) for our proposed **Transformer+Functional Embedding V2 (FUNC2)** against standard time–series baselines: **FIR** (linear causal MIMO), **TCN**, **GRU**, and **CopyBest** (correlation-selected source with scalar gain). Each dot is a subject and gray lines connect the same subject across models (paired comparison). The pooled (multi-subject) **FUNC2** model clearly outperforms all baselines, indicating that the transformer with functional coordinates can effectively *aggregate* cross-subject data to learn stronger source→target mappings. We also include a **Single-FUNC2** control (the same transformer+functional embedding trained *per subject*); as expected, it underperforms the pooled model because transformers are data-hungry and a single session in this dataset contributes only $\sim$500–900 windows—insufficient for reliable convergence. Statistical tests (paired two-sided $t$-tests vs FUNC2 with Holm and BH/FDR corrections) show all comparisons are significant ($p_{\mathrm{corr}} < 0.001$).

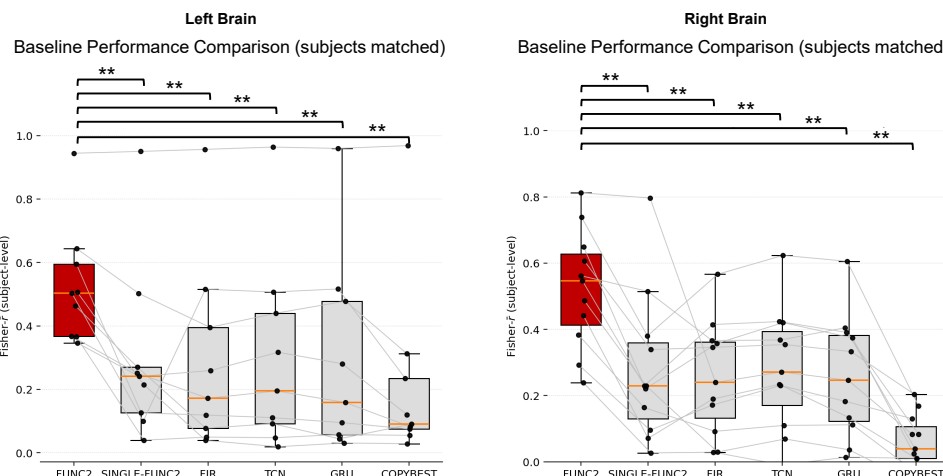

Figure A.3: *Baseline comparisons (subject-level Fisher-$\bar{r}$).* Boxes show median and IQR; dots are individual subjects; gray lines connect the same subject across models. Models: FIR (linear causal MIMO), TCN, GRU, CopyBest, and our Transformer with Functional Embedding V2 (FUNC2). The pooled (multi-subject) FUNC2 model outperforms all baselines. Paired $t$-tests vs FUNC2 with Holm and BH/FDR corrections: all $p_{\mathrm{corr}} < 0.001$. (**)

## A.7 ADDITIONAL ANALYSIS: DO FUNCTIONAL EMBEDDINGS NEED TO BE TRAINED ON EVERY SUBJECT?

To evaluate whether the functional embedding must be trained on every subject in order to support masked-region reconstruction, we performed a five-fold leave-four-subjects-out experiment separately for the right and left hemispheres. In each fold, the functional embedding was trained on 16 subjects while the remaining 4 subjects were excluded. The masked-region reconstruction transformer was then trained on *all* subjects using that fold's embedding model. Thus, every subject was "held-out" exactly once and "held-in" during the remaining folds for each hemisphere.

**Metric.** For each subject and fold, reconstruction accuracy was measured using the Fisher-$z$ corrected correlation coefficient $r$, averaged over all masked-region predictions across that subject's recording periods. Subject-level scores were aggregated across folds using an $n$-weighted Fisher mean, yielding one *held-in* and one *held-out* value per subject and hemisphere. We report the back-transformed correlation values $r$.

**Statistical comparison by hemisphere.** A paired $t$-test was used to compare held-in and held-out reconstruction accuracy within each hemisphere. For the **right hemisphere**, no significant difference was found ($p = 0.81$). The **left hemisphere** showed the same pattern, with no significant difference between held-in and held-out performance ($p = 0.703$). Across both hemispheres, held-in and held-out reconstruction scores were nearly identical, indicating that excluding individual subjects from functional-embedding training does not impair reconstruction quality for those subjects.

Results are reported in Tables A.3 and A.4 and Figure A.4.

| Subject (right) | Held-in | Held-out | $\Delta$ = held-in − held-out |
|:---:|:---:|:---:|:---:|
| 1 | 0.5533 | 0.5369 | 0.0164 |
| 2 | 0.2362 | 0.2558 | -0.0196 |
| 3 | 0.2320 | 0.2408 | -0.0088 |
| 4 | 0.5383 | 0.5355 | 0.0028 |
| 5 | 0.6583 | 0.6613 | -0.0030 |
| 6 | 0.4887 | 0.4964 | -0.0077 |
| 7 | 0.5612 | 0.5657 | -0.0045 |
| 8 | 0.3149 | 0.2934 | 0.0215 |
| 9 | 0.8520 | 0.8523 | -0.0003 |
| 10 | 0.6434 | 0.5952 | 0.0482 |
| 11 | 0.1942 | 0.1886 | 0.0056 |
| 12 | 0.3861 | 0.4520 | -0.0659 |
| 13 | 0.4103 | 0.4475 | -0.0372 |
| 14 | 0.3671 | 0.3740 | -0.0069 |
| 15 | 0.5889 | 0.5781 | 0.0108 |
| 16 | 0.2354 | 0.1874 | 0.0480 |
| 17 | 0.2890 | 0.2791 | 0.0098 |
| 18 | 0.5602 | 0.5520 | 0.0081 |
| 19 | 0.7217 | 0.7154 | 0.0064 |
| 20 | 0.8084 | 0.8049 | 0.0035 |

Table A.3: Right-hemisphere subject-level reconstruction performance when subjects were included (*held-in*) or excluded (*held-out*) from functional-embedding training. Differences are small and not systematically biased ($p = 0.81$).

| Subject (left) | Held-in | Held-out | $\Delta = $ held-in $-$ held-out |
|:---:|:---:|:---:|:---:|
| 1 | 0.4120 | 0.4293 | -0.0173 |
| 2 | 0.1816 | 0.1891 | -0.0075 |
| 3 | 0.5660 | 0.5691 | -0.0032 |
| 4 | 0.3568 | 0.3693 | -0.0125 |
| 5 | 0.4879 | 0.4998 | -0.0119 |
| 6 | 0.3660 | 0.3572 | 0.0088 |
| 7 | 0.4809 | 0.4921 | -0.0111 |
| 8 | 0.6433 | 0.6591 | -0.0158 |
| 9 | 0.5731 | 0.5761 | -0.0029 |
| 10 | 0.6495 | 0.6575 | -0.0080 |
| 11 | 0.1948 | 0.1901 | 0.0047 |
| 12 | 0.5762 | 0.5731 | 0.0031 |
| 13 | 0.2140 | 0.2326 | -0.0185 |
| 14 | 0.3702 | 0.3533 | 0.0169 |
| 15 | 0.5218 | 0.4722 | 0.0496 |
| 16 | 0.3624 | 0.3192 | 0.0433 |
| 17 | 0.4737 | 0.5297 | -0.0560 |
| 18 | 0.9316 | 0.9315 | 0.0001 |

Table A.4: Left-hemisphere subject-level reconstruction performance when subjects were included (*held-in*) or excluded (*held-out*) from functional-embedding training. As in the right hemisphere, differences are small and not systematically biased ( $p = 0.703$ ).

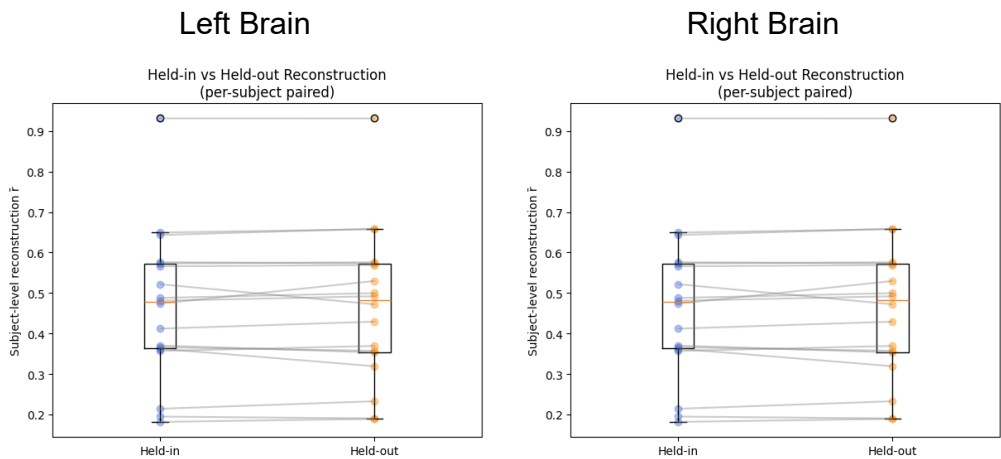

Figure A.4: *Held-in vs. held-out reconstruction performance for right and left hemispheres.* Each line connects the held-in and held-out value for a single subject. For both hemispheres, held-in and held-out distributions almost completely overlap, consistent with the non-significant differences reported in Tables A.3 and A.4.

## A.8 ADDITIONAL DECODING TASK: BAND–POWER PREDICTION

**Motivation** To further evaluate whether functional coordinates provide improved decoding performance beyond the masked–region reconstruction task, we introduce an additional decoding task with direct clinical relevance: *predicting per–window spectral band power*. Band–power is widely used in clinical neurophysiology and brain–computer interfaces because it reflects physiologically meaningful oscillatory activity (e.g., alpha, beta, and gamma rhythms). Accurate prediction of band–power from distributed neural activity therefore represents a practical and interpretable downstream application for our learned coordinate representations.

**Task Description** For each 1-second input window and each held–out target channel, the model is trained to predict the *relative band–power* of the target channel in three clinically relevant frequency ranges:

- **Alpha**: 8–13 Hz
- **Beta**: 13–30 Hz
- **Low Gamma**: 30–50 Hz

Relative band–power for a given band $b$ was defined as

$$p_b = \frac{P_b}{\sum_i P_i},$$

where $P_b$ is the absolute power in band. This normalization constrains values to the interval $[0, 1]$, improves regression stability, and aligns with common clinical practice.

**Lightweight Band–Power Transformer** Because each target channel yields only a single scalar per band (rather than a full waveform), we use a lightweight version of our transformer architecture. Specifically:

- model dimension reduced from `d_model` $= 128$ to $64$,
- number of attention heads reduced from $8$ to $4$,
- feed–forward hidden dimension reduced from $384$ to $128$,

which lowers the model size to $\sim 330$k parameters. The prediction head is a single linear layer followed by a softmax over the three bands, ensuring non–negative outputs that sum to one.

All other components, including the functional coordinate input tokens, remain unchanged. As in the main experiments, we compare two coordinate systems:

1. **MNI coordinates**: standard stereotactic coordinates.
2. **Functional coordinates (Func2)**: the learned functional embedding described in the main text.

**Evaluation** We evaluate the band–power transformer across subjects and periods separately in the *left* and *right* hemispheres. For each subject and each band, we compute the Pearson correlation coefficient $r$ between the predicted and true relative band–power across all windows in the test set. Subject-level performance is obtained by averaging $r$ across periods. Group-level statistics are computed across subjects.

**Statistical analysis.** For each band, we conduct a paired comparison between the MNI and Func2 models across subjects. Because performance values are paired and not guaranteed to be normally distributed, we report:

- a **two-sided Wilcoxon signed–rank test** ($n$-subjects), and

Positive differences (`Func2–MNI`) indicate improved performance using functional coordinates. Results are shown in Figure A.5

**Results: Left Hemisphere**  Across $n = 18$ left–hemisphere subjects, Func2 significantly outperformed MNI in all frequency bands:

**Beta (13–30 Hz).**
$$\text{mean } r_{\text{MNI}} = 0.0952, \quad \text{mean } r_{\text{Func2}} = 0.2122,$$
$$\Delta r = +0.1170, \quad \text{Wilcoxon: } p = 1.289 \times 10^{-3}.$$

**Low Gamma (30–50 Hz).**
$$\text{mean } r_{\text{MNI}} = 0.1581, \quad \text{mean } r_{\text{Func2}} = 0.3141,$$
$$\Delta r = +0.1560, \quad \text{Wilcoxon: } p = 1.907 \times 10^{-4}.$$

**Alpha (8–13 Hz).**
$$\text{mean } r_{\text{MNI}} = 0.1160, \quad \text{mean } r_{\text{Func2}} = 0.2522,$$
$$\Delta r = +0.1362, \quad \text{Wilcoxon: } p = 8.392 \times 10^{-4}.$$

**Results: Right Hemisphere**  Across $n = 20$ right–hemisphere subjects, the same pattern holds:

**Beta (13–30 Hz).**
$$\text{mean } r_{\text{MNI}} = 0.0994, \quad \text{mean } r_{\text{Func2}} = 0.2235,$$
$$\Delta r = +0.1241, \quad \text{Wilcoxon: } p = 3.153 \times 10^{-3}.$$

**Low Gamma (30–50 Hz).**
$$\text{mean } r_{\text{MNI}} = 0.1933, \quad \text{mean } r_{\text{Func2}} = 0.3192,$$
$$\Delta r = +0.1260, \quad \text{Wilcoxon: } p = 3.654 \times 10^{-3}.$$

**Alpha (8–13 Hz).**
$$\text{mean } r_{\text{MNI}} = 0.1229, \quad \text{mean } r_{\text{Func2}} = 0.2574,$$
$$\Delta r = +0.1345, \quad \text{Wilcoxon: } p = 2.712 \times 10^{-3}.$$

**Summary**  Across both hemispheres and all tested frequency bands, functional coordinates consistently yield substantially higher correlation between predicted and true band power. These results demonstrate that the geometry learned by the functional coordinate system supports meaningful and clinically relevant decoding tasks.

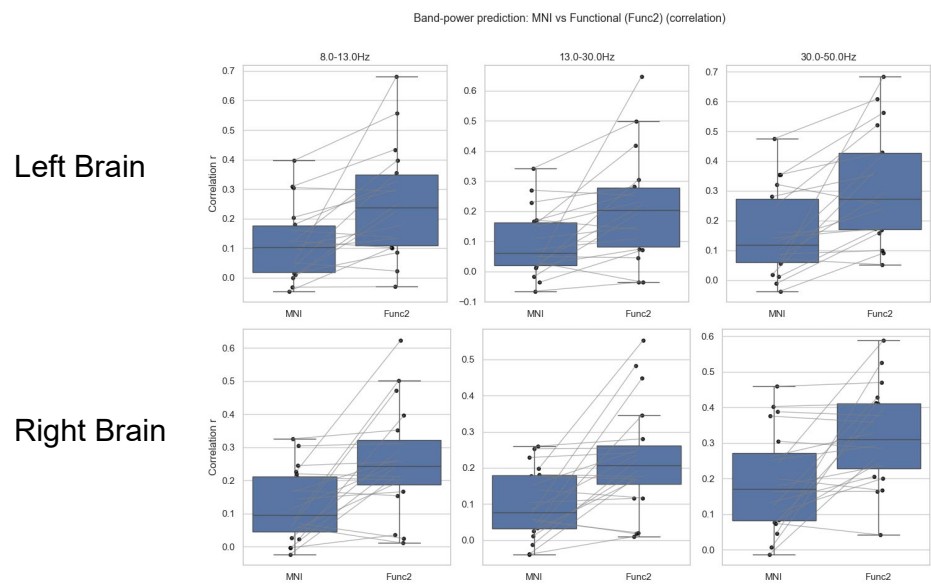

Figure A.5: **Band–power decoding performance comparing MNI vs. functional coordinates.** For each frequency band (columns) and each hemisphere (top: left, bottom: right), we show subject-level boxplots of the prediction correlation $r$, along with paired data points and connecting lines indicating within-subject differences. In all bands and both hemispheres, the functional coordinate model (`Func2`) significantly outperforms the MNI baseline.

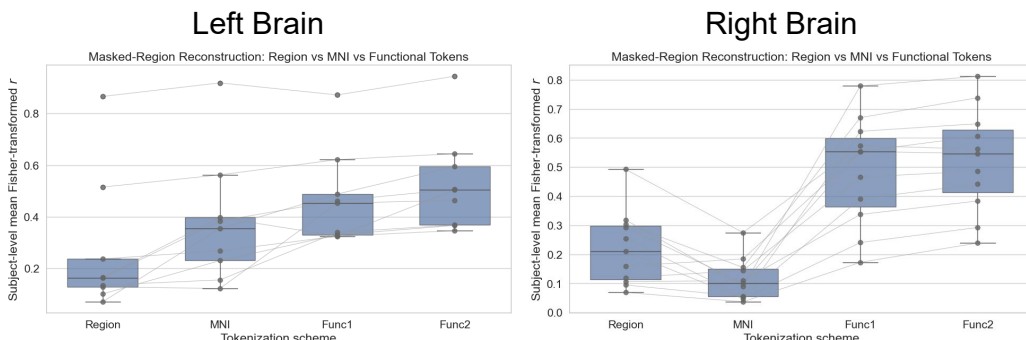

Figure A.6: **Functional embeddings outperform region-label tokenization.** Subject-level Fisher-transformed reconstruction correlations ($r$) for the Region-token Transformer (one-hot region encoding) and the Functional Transformer (Func2), shown for left and right hemispheres. Gray points denote individual subjects; lines connect the same subject across tokenization schemes. In both hemispheres, Func2 yields significantly higher reconstruction accuracy than Region (left: $p = 1.953 \times 10^{-3}$; right: $p = 4.883 \times 10^{-4}$; Wilcoxon signed-rank test), demonstrating that functional embeddings capture structure beyond coarse anatomical region labels.

### A.9 ADDITIONAL CONTROL: ARE FUNCTIONAL EMBEDDINGS SIMPLY ENCODING REGION LABELS?

A potential question is that the performance advantage of the Functional Transformer may arise because the functional encoder is implicitly leveraging coarse region labels (e.g., GPi, STN, VO) rather than learning genuinely functional structure. Since the contrastive training stage has access to channel identity information, including anatomical region, it is important to test whether the observed gains can be reproduced simply by providing the model with region labels directly, without any functional encoder.

**Region-token Transformer.** To address this concern, we constructed a control model in which channel identity is encoded solely through a one-hot region label of matching dimensionality. This *Region-token Transformer* uses the same architecture, masked-region reconstruction objective, training schedule, dataset, and data splits as the Functional Transformer. The only difference is the tokenization scheme: instead of using functional embeddings learned through contrastive learning (Func2), channels are represented by their anatomical region labels alone.

**Comparison against functional embeddings.** We evaluated masked-region reconstruction performance across subjects for both hemispheres using Fisher-transformed reconstruction correlations ($r$). Each subject contributes a paired measurement: one from the Region-token model and one from Func2. This allows a direct assessment of whether functional embeddings provide value beyond region identity.

**Results.** The comparison (Fig. A.6) shows a clear and systematic reduction in reconstruction accuracy when the model uses only region identity. Importantly, this reduction is statistically significant in both hemispheres:

- **Left hemisphere:** Func2 > Region    ($p = 1.953 \times 10^{-3}$, Wilcoxon signed-rank).
- **Right hemisphere:** Func2 > Region    ($p = 4.883 \times 10^{-4}$, Wilcoxon signed-rank).

**Conclusion.** These results show that providing only anatomical region identity is insufficient to reproduce the performance gains achieved via functional embeddings. The functional encoder therefore learns structure that is not reducible to region labels, supporting our claim that the functional representation captures data-drive and meaningful functional organization that improves downstream masked-region reconstruction, pure over anatomical coordinates (MNI or region labels).

## A.10   SCALING ANALYSIS: EFFECT OF DATASET SIZE ON MASKED-REGION RECONSTRUCTION

To assess how the amount of training data influences the performance of the masked-region reconstruction model, we performed a scaling analysis in which the size of the *Functional Transformer* training set was systematically varied. Importantly, the functional encoder used to generate the channel embeddings was kept fixed throughout this analysis. Only the number of subjects contributing data to the masked-region reconstruction training procedure was varied.

We trained a series of Functional Transformer (Func2) models using progressively larger subsets of subjects drawn from our full dataset: $\{3, 5, 8, 10, 13, 15, 18, 20\}$ subjects. For each subset size, we trained a separate transformer on the same masked-region reconstruction objective described in the main text. All models were evaluated on the test sets, ensuring a consistent and comparable evaluation across all dataset sizes.

**Evaluation metric.**   For each model, we computed Fisher-transformed reconstruction correlations ($r$) on the full test set and report the Fisher-mean $\bar{r}$ across all masked segments. Because both the functional encoder and the test set are identical across conditions, the resulting changes in performance can be attributed solely to differences in the amount of cross-subject training data available to the masked-region reconstruction transformer.

**Results.**   Figure A.7 shows the relationship between the number of subjects included in the training data and the resulting reconstruction performance. We observe a clear improvement as the dataset grows from 3 to 5 subjects, followed by a local plateau around 8 subjects. Interestingly, increasing the dataset size beyond 10 subjects leads to a renewed and substantial gain in reconstruction accuracy, with the highest performance obtained when all 20 subjects are included.

This pattern suggests that while a moderate number of subjects ($\sim$8) is sufficient to achieve stable mid-range performance, **incorporating additional subjects continues to produce measurable improvements**. These gains likely reflect the benefit of aggregating neural recordings across a diverse set of individuals, enabling the model to learn cross-subject regularities and region-to-region dependencies that do not appear in smaller datasets.

## A.11   EXTENDED RESULTS:SINGLE SUBJECT FUNCTIONAL EMBEDDING AND CONFUSION MATRIX

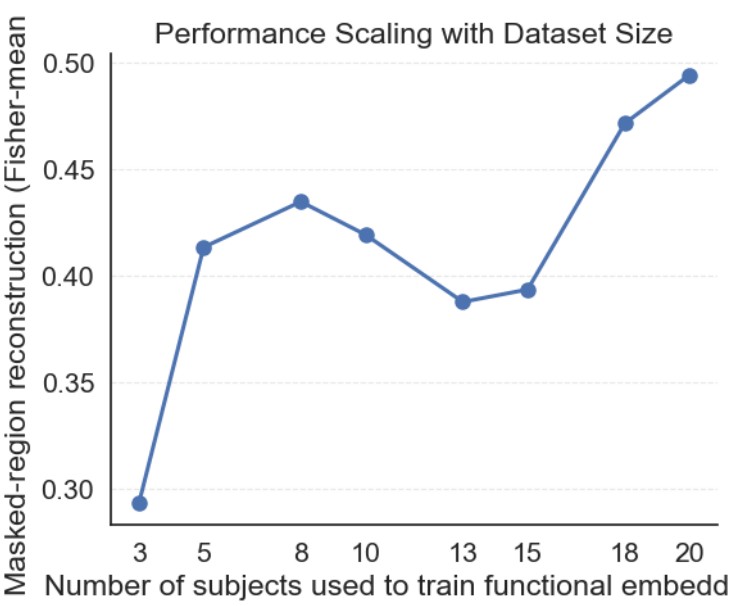

Figure A.7: *Impact of dataset size on masked-region reconstruction.* Fisher-mean reconstruction correlation on a fixed test set (15% of every recording session) for Functional Transformer models trained with data from $\{3, 5, 8, 10, 13, 15, 18, 20\}$ subjects. An early plateau is visible near 8 subjects, but performance improves again with larger datasets, reaching its peak when all 20 subjects are included. Since the functional encoder is fixed, these improvements directly reflect the benefit of scaling cross-subject training data for the reconstruction transformer.

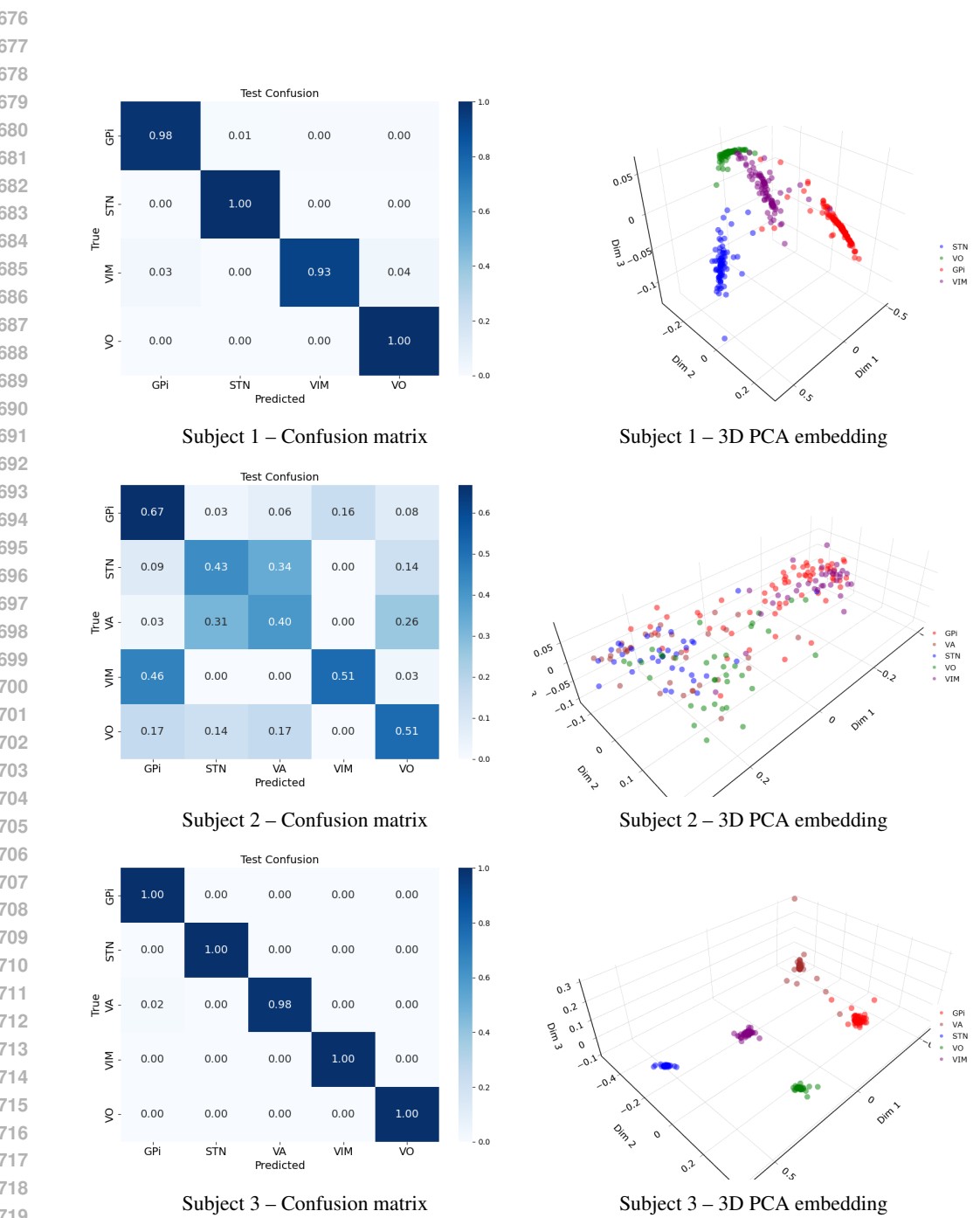

Figure A.8: Per-subject confusion matrices and 3D PCA functional embeddings for held-out time segments. Each row corresponds to one subject: left panel shows the classification confusion matrix; right panel shows the 3D PCA projection of the learned functional embedding. For each subject a random example period is shown.

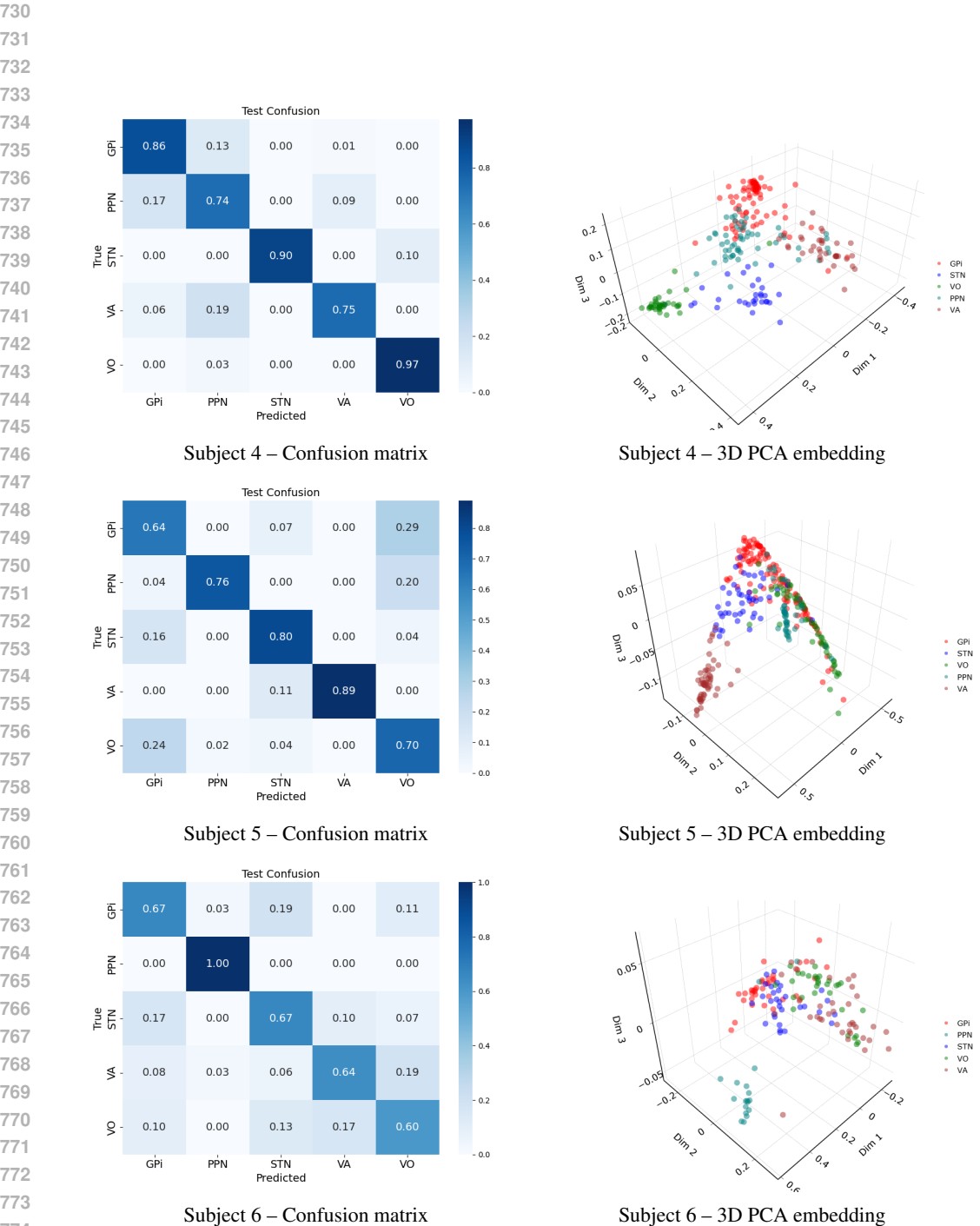

Figure A.8: Per-subject confusion matrices and 3D PCA functional embeddings for held-out time segments (continued).

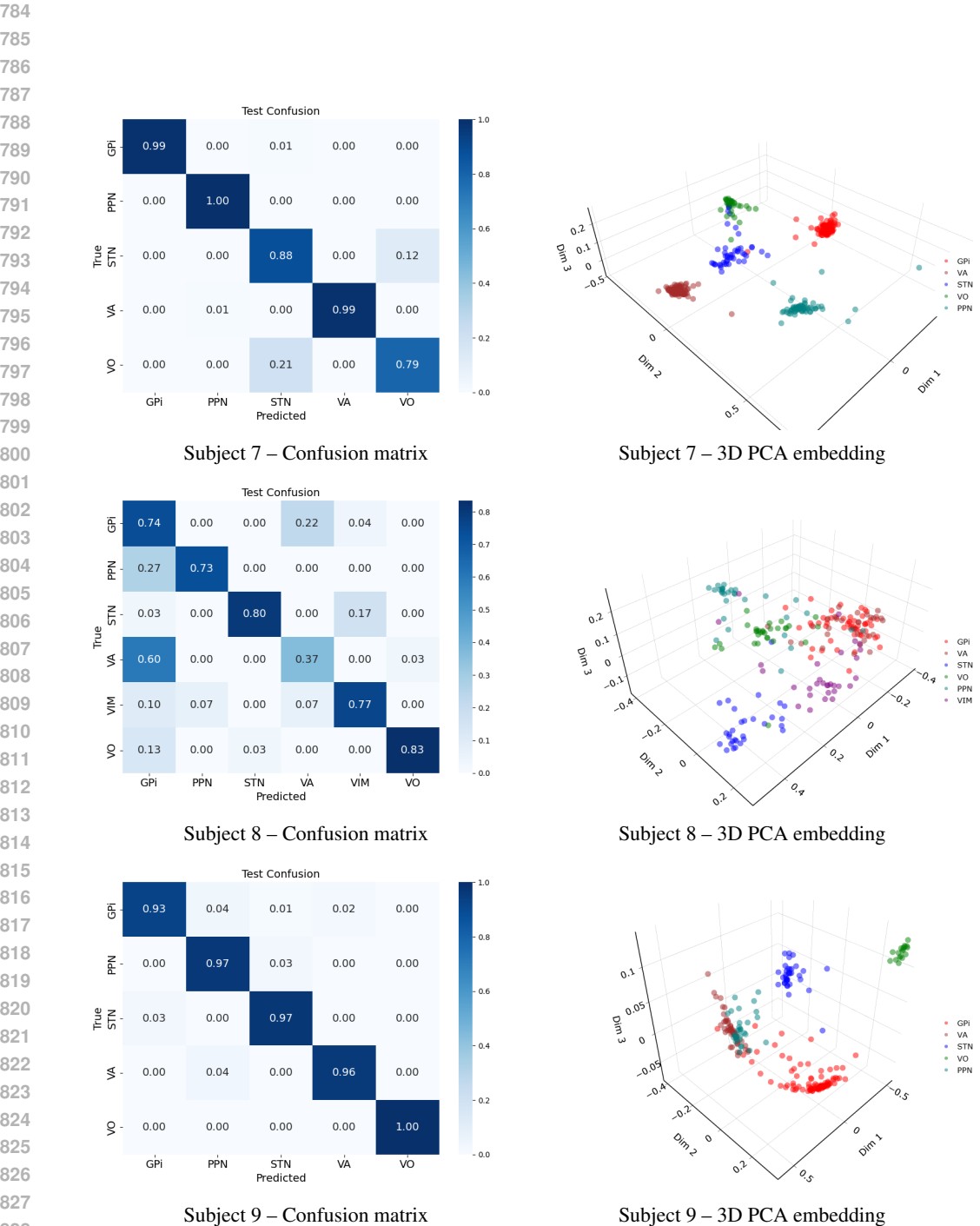

Subject 7 – Confusion matrix    Subject 7 – 3D PCA embedding

Subject 8 – Confusion matrix    Subject 8 – 3D PCA embedding

Subject 9 – Confusion matrix    Subject 9 – 3D PCA embedding

Figure A.8: Per-subject confusion matrices and 3D PCA functional embeddings for held-out time segments (continued).

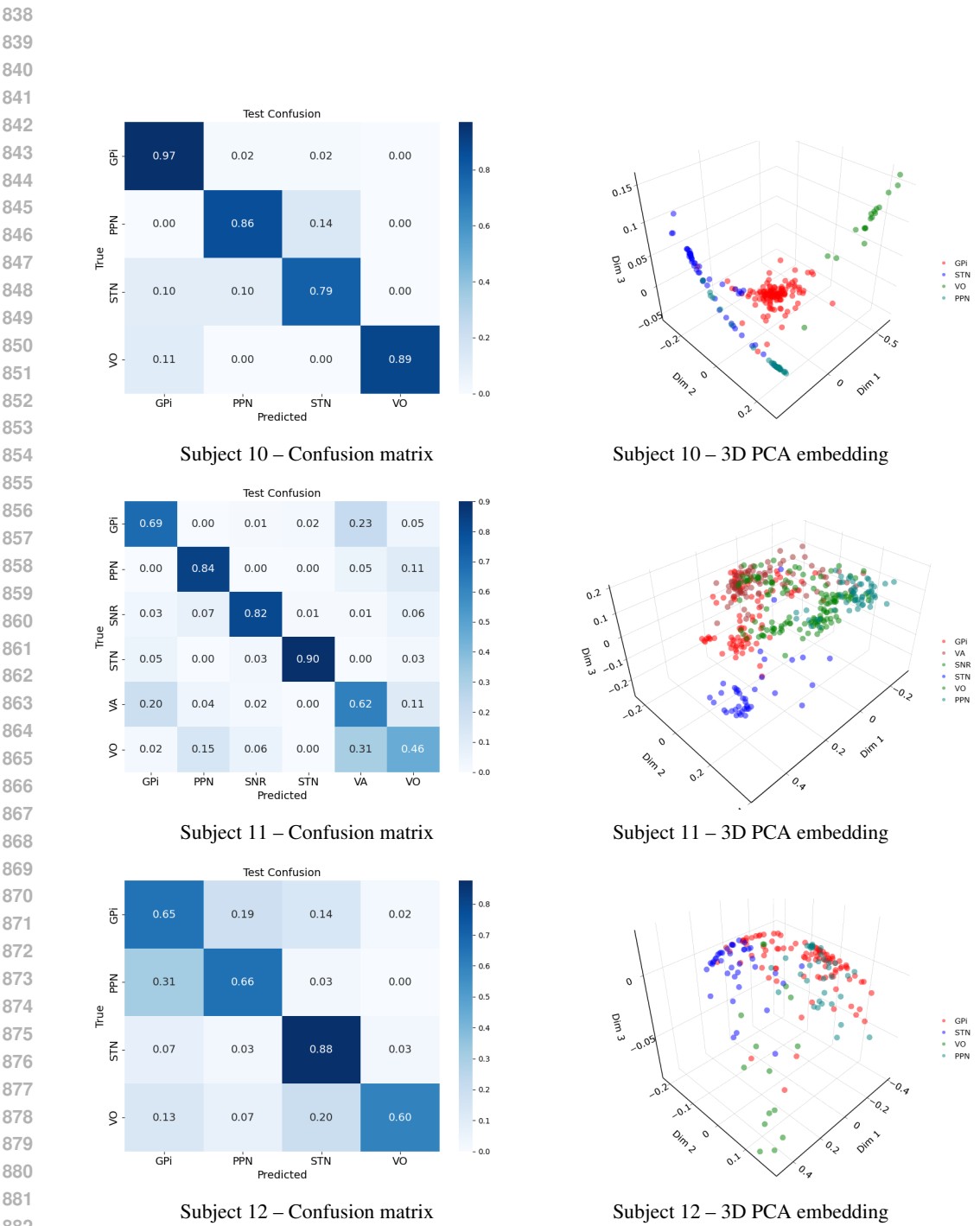

Subject 10 – Confusion matrix

Subject 10 – 3D PCA embedding

Subject 11 – Confusion matrix

Subject 11 – 3D PCA embedding

Subject 12 – Confusion matrix

Subject 12 – 3D PCA embedding

Figure A.8: Per-subject confusion matrices and 3D PCA functional embeddings for held-out time segments (continued).

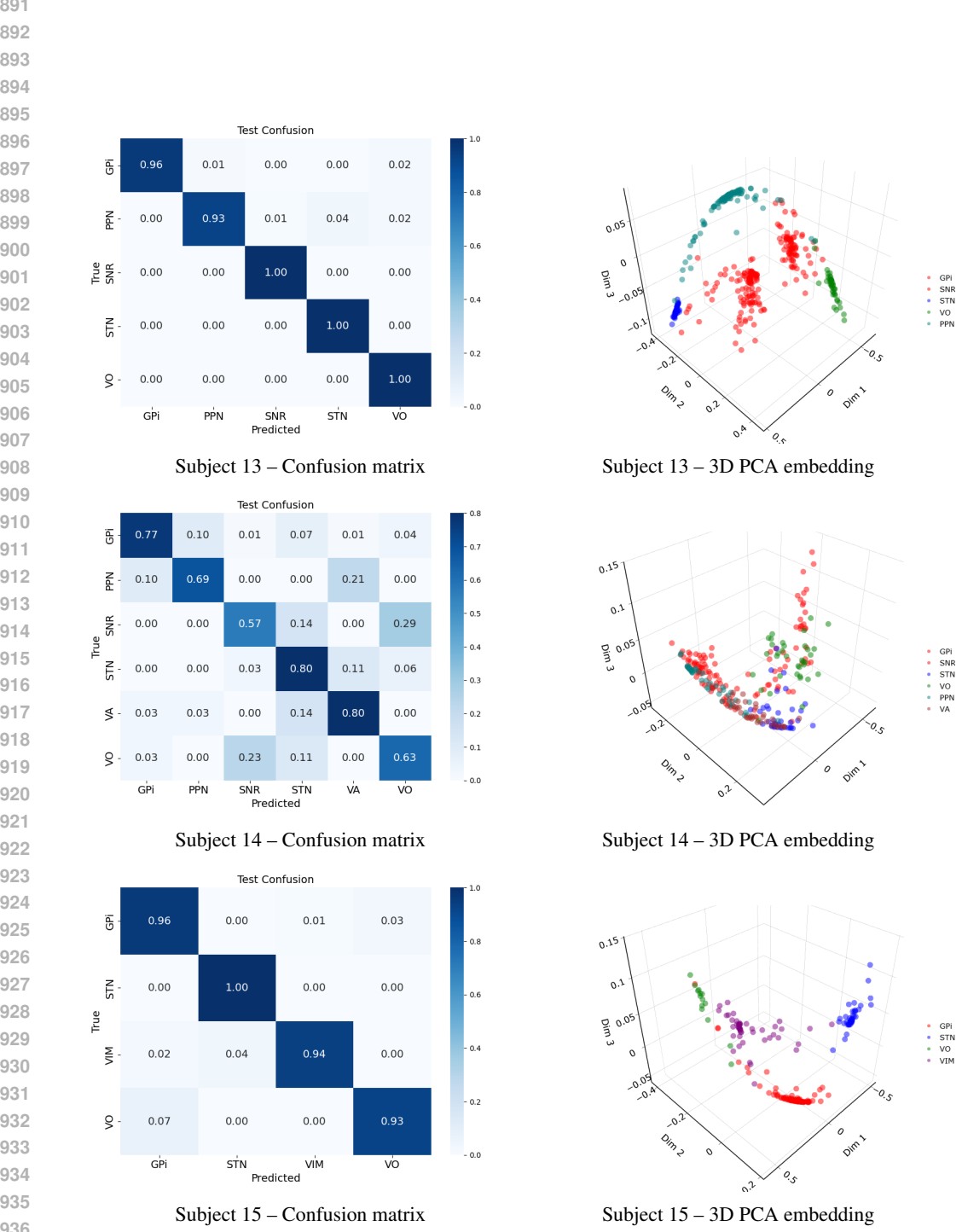

Subject 13 – Confusion matrix

Subject 13 – 3D PCA embedding

Subject 14 – Confusion matrix

Subject 14 – 3D PCA embedding

Subject 15 – Confusion matrix

Subject 15 – 3D PCA embedding

Figure A.8: Per-subject confusion matrices and 3D PCA functional embeddings for held-out time segments (continued).

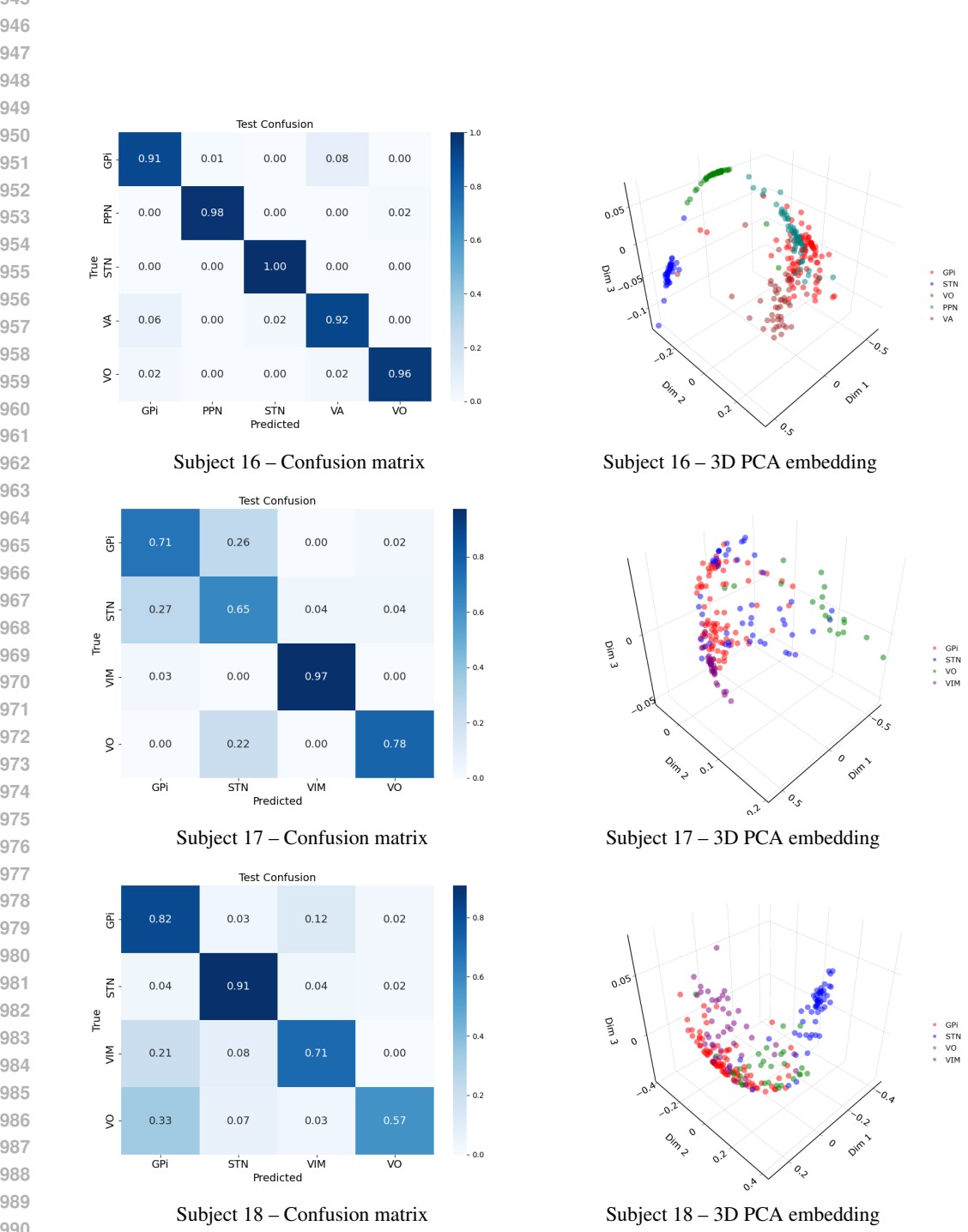

Subject 16 – Confusion matrix

Subject 16 – 3D PCA embedding

Subject 17 – Confusion matrix

Subject 17 – 3D PCA embedding

Subject 18 – Confusion matrix

Subject 18 – 3D PCA embedding

Figure A.8: Per-subject confusion matrices and 3D PCA functional embeddings for held-out time segments (continued).

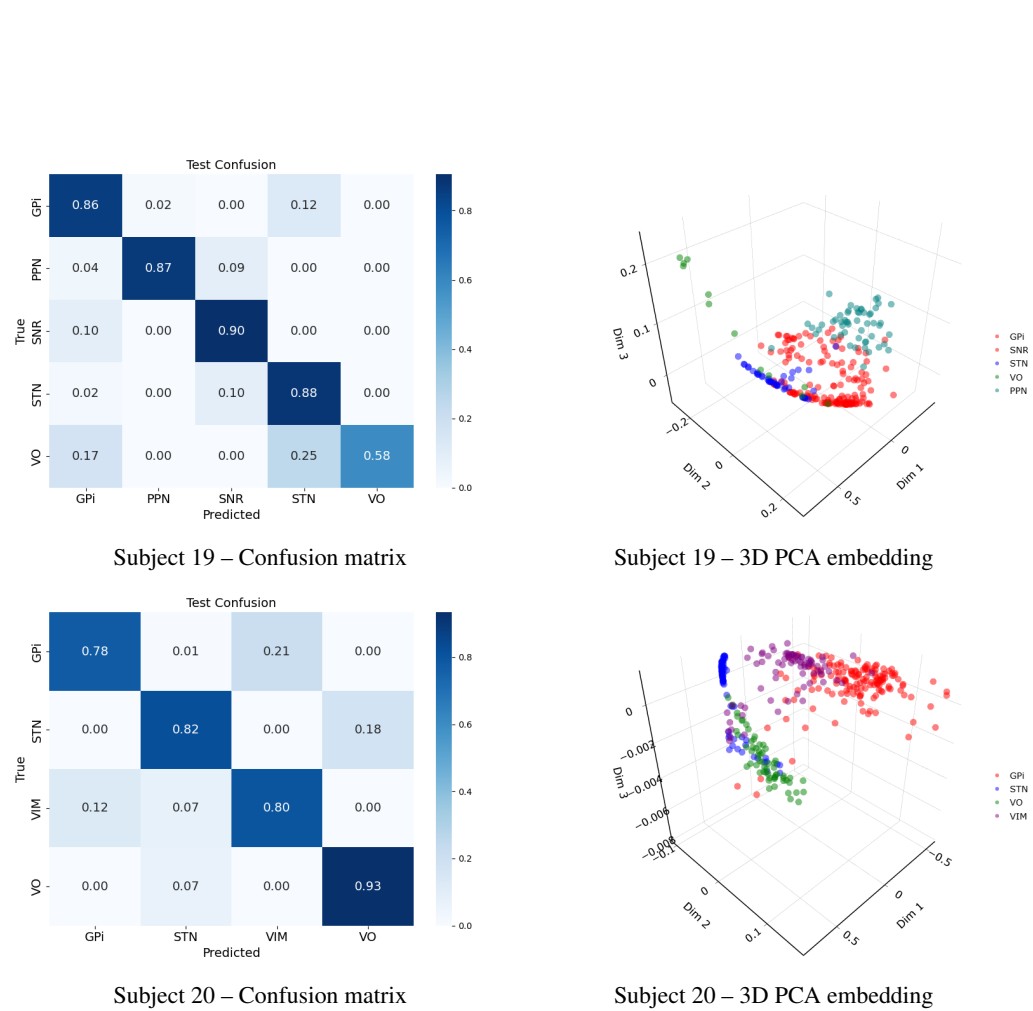

Subject 19 – Confusion matrix

Subject 19 – 3D PCA embedding

Subject 20 – Confusion matrix

Subject 20 – 3D PCA embedding

Figure A.8: Per-subject confusion matrices and 3D PCA functional embeddings for held-out time segments (continued).

## A.12 COMPUTE RESOURCES AND TRAINING TIMES

All models were trained on a single workstation equipped with an AMD Ryzen Threadripper PRO 7955WX (16 cores), 128 GiB RAM, and $1\times$ NVIDIA RTX 4500 Ada (24 GiB VRAM). Table A.5 summarizes training times and data scale for each model. Notably, the *Functional encoder (PSC, multi-subject)* is pair-based: given $n$ input samples, it can form $\mathcal{O}(n^2)$ pairs, enabling large-scale training (here, 2,500,000 pairs). In contrast, *Functional encoder (MSC, multi-subject)* computes its loss over full sample batches (not pairwise), so the effective sample count remains at 65,000 despite the same source dataset.

Table A.5: *Training times and data scale.* Reported on a single workstation (AMD Ryzen Threadripper PRO 7955WX, 128 GiB RAM, $1\times$ NVIDIA RTX 4500 Ada 24 GiB).

| Model | Epochs | Samples | Time |
|---|---|---|---|
| Functional encoder (single-subject) | 100 | 20,000 | $\sim$7 minutes |
| Functional encoder (PSC, multi-subject) | 30 | 2,500,000 | $\sim$5 hours |
| Functional encoder (MSC, multi-subject) | 200 | 65,000 | $\sim$1.5 hours |
| Masked reconstruction transformer | 150 | 27,000 | $\sim$2.5 hours |

## A.13 LARGE LANGUAGE MODELS (LLMS) USAGE

We used ChatGPT (OpenAI GPT-5) as a general-purpose assistant during the preparation of this work. The model was employed to help with editing for clarity, conciseness, and formatting. All scientific ideas, analyses, experiments, and interpretations presented in this paper were conceived, implemented, and validated by the authors. The authors take full responsibility for the content of this manuscript.

