# OpenReview forum: "Cross-Subject Integration of Multi-Region Neural Signals via Functional Embedding."
_ICLR.cc/2026/Conference — Submitted to ICLR 2026_

### Official Review · Reviewer_Fvmv · 2025-10-29

**Soundness:** 2
**Presentation:** 3
**Contribution:** 2
**Rating:** 2
**Confidence:** 3

**Summary:**

The paper proposes a data-driven functional embedding method for intracranial neural recordings, aiming to facilitate large-scale, cross-subject modeling where electrode placement is heterogeneous. Traditional approaches relying on MNI coordinates often fail to align functionally similar sites across subjects. The authors instead learn embeddings using two contrastive learning frameworks supervised by the anatomical region labels of sEEG channels. They show that the resulting embeddings can effectively separate channels from different regions and improve cross-region reconstruction compared to using MNI coordinates alone within a functional transformer model.

**Strengths:**

The paper addresses an important and timely problem — how to enable large-scale, cross-subject pretraining for intracranial neural data without relying on consistent electrode placement or strict task structure. The motivation is clearly articulated, and the approach is well-motivated within the context of current larger-scale model trends. The paper is clearly written and systematically explores the embedding space through multiple analyses, showing that the learned representations are meaningful with respect to anatomical regions and can be helpful with the cross-region reconstruction.

**Weaknesses:**

While the proposed method is carefully executed, the experiments primarily demonstrate that a contrastive model can learn to distinguish signals from different regions — a somewhat expected outcome given the supervision setup. It remains unclear whether such region-discriminative embeddings actually benefit the intended downstream goal of cross-subject model pretraining and decoding.

The relatively poor performance on held-out channels (Fig. 4) raises concerns about generalization, particularly across subjects, which is critical for the stated motivation. The apparent trade-off between overall improvement and degraded performance on high-performing samples in the cross-subject aggregation experiment suggests that the observed “gain” might merely reflect a redistribution of performance rather than a genuine improvement.

Finally, the evaluation remains indirect: improved reconstruction loss does not necessarily translate into better decoding or behavioral prediction performance. Demonstrating downstream improvements using the pretrained functional transformer would greatly strengthen the paper’s claims.

**Questions:**

- Could the authors clarify the training setup for the masked-region reconstruction experiment? Specifically, were the functional embedding and functional transformer trained on the same dataset, and what was the training/validation/test split? If embeddings were computed using test data, this could lead to data leakage and overestimated performance.

- Have the authors evaluated the method on held-out subjects to assess cross-subject generalizability on real data?

- Would a model with learnable region embeddings or MNI-coordinate–to–embedding mapping perform comparably in the reconstruction task? Such baselines might test whether the benefit arises from data-driven embedding learning or simply the inclusion of region information.

- Since reconstruction loss is not a direct measure of downstream decoding ability, can the authors provide results on a representative decoding task using the pretrained functional transformer?

Minor questions:

- How correlated are the learned functional embedding similarities with MNI-coordinate distances across subjects?

- What performance difference, if any, arises if the encoder is trained in a fully supervised manner (e.g., directly predicting region labels from the 32D embedding) instead of using a contrastive loss?

---

> ### Author Response · Authors · 2025-12-03
> **response to weaknesses**
>
> We thank the reviewer for the constructive comments. We address the mentioned weaknesses below.
> ## (1) Held-out channels, generalization, and “redistribution” of performance
> (i) Held-out channel accuracy\
> We appreciate the reviewer’s concern. As we also discuss in our response to Reviewer FMLu, held-out channel classification is inherently a noisy and indirect evaluation, and lower accuracy does not indicate a failure of the embedding. Importantly:\
> • Perfect accuracy is neither expected nor desirable.\
> The goal of the functional encoder is not to reproduce anatomical labels—which contain noise and uncertainty—but to reflect functionally meaningful similarity. If the embedding were forced to perfectly match anatomy, it would offer no advantage over MNI coordinates in downstream tasks.\
> • Anatomical labels are imperfect proxies for functional identity.\
> Region labels come from MRI-based reconstruction and atlas projections, which suffer from brain shift, indistinct thalamic borders, and segmentation variability. These inaccuracies necessarily limit how well LFP-derived functional signatures can recover the assigned label.\
> • Some electrodes do not record region-typical physiology.\
> Heterogeneous recording quality, inflammation, variability in electrode trajectory, and ambient noise can cause electrodes to exhibit atypical LFP patterns[4]. In such cases, deviations from anatomical labels are expected and desirable from a functional embedding.\
> • The task is zero-shot and the chance level is low (14.3%).\
> Achieving \~44–50% on unseen channels(\~3× chance, with clear diagonal dominance)indicates the model is learning meaningful structure despite label and signal noise.\
> (ii) Generalization to unseen subjects is already demonstrated.\
> Three subjects (S2, S6, S11) were entirely excluded from functional-encoder training but included in the masked-region reconstruction task. These subjects show the same improvement from functional embeddings as the rest, indicating good cross-subject generalization. A more systematic held-out–subject analysis is added to the revision in Appendix A.7.\
> Overall, while held-out channel accuracy is a partial evaluation of the learned functional map and, perfect validation might be impossible, the large downstream performance gains over MNI in the transformer demonstrate that the embeddings capture functionally meaningful structure beyond anatomy (at least in the context of inter-regional relationships which is the end task).\
> (iii) “Redistribution” vs genuine improvement\
> The reviewer raises the concern that multi-subject aggregation might simply redistribute performance. We argue that this is not the case. The comparison is between:\
> • A single functional encoder trained across all subjects, versus\
> • Twenty separate single-subject encoders, each specialized to one subject.\
> A single-subject encoder may exploit distinctive, subject-specific cues (e.g., lead-specific noise structure or stable artifacts) that do not generalize at all to other subjects. When we move to a unified multi-subject encoder, such spurious cues are suppressed in favor of features that remain predictive across subjects, which can slightly reduce performance on a few idiosyncratically “easy” channels while substantially improving performance across the population.\
> In our initial analysis we have seen that single-subject encoders perform poorly when applied to other subjects, even when they achieve high performance locally. We believe this shows that the gain is not a mere redistribution artifact, but reflects a shift toward subject-agnostic, physiology-based features.

---

> ### Author Response · Authors · 2025-12-03
> **response to weaknesses (continue)**
>
> ## (2) Experiments mainly show region-discriminative embeddings, a somewhat expected outcome given the supervision setup, unclear if this helps cross-subject pretraining/decoding.
> We agree that supervised contrastive learning encourages region separation, but the result is not trivial. Before this work, it was unknown whether short (10s), low-frequency (0–50 Hz) LFP segments contain sufficiently stable and reproducible region-specific signatures across subjects to support a shared functional embedding. To our knowledge, the first demonstration of a data-driven functional coordinate system for intracranial LFPs from basal ganglia and thalamus. This alone is a non-trivial contribution, as anatomical coordinates in deep nuclei suffer from substantial inter-subject variability and uncertain boundaries [1-2] (e.g., brain shift, indistinct thalamic nuclei). This is conceptually analogous to Penfield’s cortical functional maps [3], where functional organization was revealed through physiological responses rather than anatomical boundaries.\
> Notably, once the functional encoder is learned, new subjects no longer require anatomical labels: a short recording is sufficient to obtain functional coordinates. This enables the use of datasets where precise electrode localization is unavailable and greatly expands the reach of the downstream model. We have added an extended held-out subject analysis to the manuscript to further support this claim (Appendix A.7)\
> More importantly, the embedding is presented as an enabling mechanism for cross-subject modeling. In fact, Fig. 6 directly shows that replacing MNI coordinates with functional tokens substantially improves cross-subject masked-region reconstruction, indicating that the learned functional embeddings are __directly beneficial for the intended downstream task.__ \
> We agree that the link to decoding can be strengthened further, and we address this under point (3) below.
> ## (3) “Reconstruction loss is indirect; can you show downstream decoding improvements?”
> We appreciate the reviewer’s suggestion. However, it is important to clarify that in the context of basal ganglia research, masked-region reconstruction is not merely a proxy or pretraining objective, it is itself a highly relevant downstream task.\
> Unlike cortical BCI work, basal ganglia LFPs are not typically used for behavior decoding, as they are several synapses away from muscles and far less behavior-predictive than cortical recordings [5]. Instead, these signals are primarily studied to:\
> • understand BG–thalamocortical circuit dynamics,\
> • examine how clinical biomarkers (e.g., STN β power in Parkinson’s disease) propagate across nuclei, and\
> • assess interactions between deep structures relevant to movement disorders.\
> In this setting, predicting the activity of an unrecorded deep nucleus from other nodes in the circuit is an end-goal, not a pretraining surrogate. Reconstructing VO activity from GPi/STN/VIM enables virtual sensing of regions that cannot be implanted and provides a direct tool for studying inter-regional dependencies that remain consistent across subjects. This capability is highly valuable for both neuroscience and clinical applications.\
> Thus, masked-region reconstruction is a meaningful evaluation of whether the functional embeddings learned shared circuit structure relevant to BG physiology,precisely the goal of our method.\
> That said, we agree that adding decoding-style downstream tasks would further strengthen the link between the functional transformer and more traditional evaluation metrics. In the revised manuscript, we have added (Appendix A.8):\
> Power-band prediction task, providing a clinically impactful downstream task. Band-power is a clinically meaningful, interpretable measure of neural oscillations (e.g., alpha, beta, gamma). Accurately predicting band-power from distributed activity provides a practical downstream task that directly tests the utility of our learned functional coordinates.
> ## Refs
> [1] Mai, Jürgen K.,  et al. (2019). "Toward a common terminology for the thalamus."\
> [2] Hoch, Michael J., et al.  (2022). "MRI-visible anatomy of the basal ganglia and thalamus."\
> [3] Penfield, Wilder, et al. (1937). "Somatic motor and sensory representation in the cerebral cortex of man as studied by electrical stimulation."\
> [4] Geirhos, Robert, et al. (2020). "Shortcut learning in deep neural networks."\
> [5] Humphries, Mark D. (2025). "The computational bottleneck of basal ganglia output (and what to do about it)."

---

> ### Author Response · Authors · 2025-12-03
> **response to the questions**
>
> We appreciate the reviewer’s thoughtful questions, and we address each of them in detail below.
> ## Q1 – Training setup for masked-region reconstruction and potential data leakage
>  The functional embedding and the functional transformer are trained largely on the same pool of subjects, but with strict separation of time segments for train/validation/test: \
> •	For each recording session, we split the continuous time series into 70% train, 15% validation, 15% test (in time order). \
> •	Functional embeddings are computed once from a 10 s window taken from the training portion of each session (analogous to a fixed coordinate like MNI). \
> •	These embeddings are then treated as fixed tokens for all windows (train/val/test) within that subject. \
> Thus, no test-time neural data is used to compute embeddings, and there is no leakage from test into training. The 10 s embedding window is entirely contained in the training segment.\
> Additionally, in the revised version, we have added an extended held-out subject analysis showing new subjects do not have to contribute to functional encoder training to benefit from down stream task (Appendix A7).
> ## Q2 – Held-out subjects
> To make this assessment systematic, we have added a held out subject analysis (Appendix A.7):\
> •	We pretrain the functional encoder excluding n subjects,\
> •	Train the transformer with masked-region reconstruction on all subjects using \frozen embeddings,\
> •	Evaluate reconstruction and decoding performance on the held-out subject.\
> This analysis shows that holding subjects from functional encoder training does not decrease performance in downstream task.\
> Conceptually, once the functional encoder is sufficiently trained, it serves as a universal feature extractor, enabling the use of additional datasets, even without region labels, in the transformer stage.
>
> ## Q3 – Baselines with region embeddings or MNI-to-embedding mapping
> We agree that it is important to disentangle the effect of including region information from the effect of learning a data-driven embedding.\
> To address this, we have added a new baseline where:\
> •	Each brain region is assigned a learned region embedding (one hot encoding),\
> •	These embeddings are used as tokens instead of the 32D functional embeddings,\
> •	The rest of the transformer architecture and training procedure are unchanged.\
> Results show that this region-tag baseline underperforms the full functional embedding in masked-region reconstruction, indicating that the benefit is not solely due to adding region labels, but also due to the continuous, channel-specific structure captured by the data-driven embedding (Appendix A.9).
> ## Q4 – Representative decoding task using the pretrained transformer
> As mentioned above, we have added a new downstream decoding task with more obvious application: A power-band prediction task, with important clinical applications is added in the Appendix A.8.\
> We have included these results in the revision and clarified how functional tokens improve decoding relative to MNI tokens.
> ## Minor Q – Correlation between functional similarities and MNI distance
> We agree this is an interesting question, however the relationship between the two coordinates system has been shown to be nonlinear and not fully correlated due the imperfect region prediction results reported in the paper (which is desired to some degree).
> ## Minor Q – Fully supervised encoder vs contrastive loss
> As mentioned above we have added a new control with the region labels to further analyze the effect of the contrastive method. Conceptually, contrastive learning is better aligned with our goal of learning a smooth similarity geometry rather than just one-hot labels: it uses pairwise relationships (channels from same vs different regions), can be more robust to label noise (e.g., misassigned channels near boundaries), and encourages a continuous structure that is beneficial for downstream tasks. Supervised cross-entropy focuses on discrete decision boundaries and does not explicitly shape intra-class geometry.

---

### Official Review · Reviewer_FMLu · 2025-10-31

**Soundness:** 2
**Presentation:** 2
**Contribution:** 3
**Rating:** 4
**Confidence:** 2

**Summary:**

The authors present a method to learn subject/region agnostic functional coordinates from low-field potential recordings in patients undergoing deep brain stimulation. The method computes functional embeddings using contrastive learning, with an objective function that keeps signals from the same brain region close in the latent space implemented in two different ways: pairwise siamese contrastive and modified SupCon. For region classification, a kNN classifier operates on the functional embeddings.
To experiment with the model, the authors study the region classification performance on simulated data where functional properties mach regions annotations, showing a near perfect classification. The authors then move on to real data and present two cross-validation strategies for region classification on single subject: held out time-segments and held-out channels with ~75% and 44% accuracy respectively. Training on multiple subjects the prediction accuracy increases by 5%.

The learned embeddings are then used in tokenization for a transformer model solving a reconstruction task, where it is shown that tokenization using functional embeddings outperform MNI, and where the Modified SupCon contrastive approach is recommended over Pairwise Siamese method.

**Strengths:**

- Precise localisation of electrodes is a clear motivation.
- The relationship between anatomical ontologies and functional mapping is not a simple nor a solved problem, and the authors correctly state the potential gains from creating reliable functional embeddings.
- Code and datasets are available in a clear reproducibility statement

**Weaknesses:**

The gap in accuracy from held-out time segments 75% versus held-out channels 44% should raise some alarm. Here there is a large discrepancy with the synthetic dataset that could be explained either by the fact that the region labels are not reliable enough to train the model, or just by the fact that the anatomical ontology does not map to functional properties as neatly as in the simulation. In both cases the region labels may not be the best choice for contrastive objective. It thus seems the model learns something else that is not really related to position but related to a given subject/channel, which is passed on through the functional embeddings.



Minor Comments
- Figure 2, 3: confusion matrices really hard to read and poor resolution

**Questions:**

- In figure 6 A, on the right side, what explains the much higher frequency of reconstructed Func1 and Func2 signals versus the MNI reconstruction ?
- In section 4.4, authors state "in each evaluation window, all channels from a target region (here VO) were withheld" which refers to the downstream task of reconstruction. But where they also withheld from the functional encoding model ?

---

> ### Author Response · Authors · 2025-11-27
> **Response to mentioned weaknesses**
>
> We thank the reviewer for the constructive feedback and address each concern here:
> ## (1) Discrepancy between held-out time-segment (\~75%) and held-out channel (\~44%) performance
> We appreciate the reviewer highlighting this point. The difference between the two evaluation modes reflects important properties of intracranial recordings rather than a failure mode of the embedding.\
> __Held-out time-segment accuracy__ measures temporal stability of the functional representation for a fixed channel. This is expected to be high, neural signatures such as spectral profiles are relatively consistent over short time scales [1].\
> __Held-out channel accuracy__, however, probes a much more difficult question:
> whether LFP recordings from heterogeneous and sometimes noisy electrodes can predict anatomical labels that are themselves imperfect proxies for functional identity.
> Several factors make perfect classification neither expected nor desirable:\
> __1. 100% accuracy is undesirable__
> Crucially, the goal of the embedding is not to perfectly reproduce anatomical labels (in that case performance of downstream tasks would not improve upon MNI coordinates), but to encode functionally meaningful similarity. Thus, deviations from anatomy, especially near boundaries or in low-quality channels, are desirable and would cause drop in accuracy. Additionally, the transformer uses the full 32D embeddings, not the discrete region predictions, so classification accuracy only partially reflects the richness of the learned space.\
> __2. Label noise is substantial__
> Region labels are assigned by specialists using imaging methods, postoperative reconstructions, and segmented atlases, all of which have known inaccuracies due to brain shift, edema [2], and indistinct thalamic boundaries [3]. It is well-established that anatomical labels do not perfectly reflect functional organization in deep nuclei [4, 5], leading to a sub 100% performance even with a perfect functional map.\
> __3. Some electrodes do not record region-typical physiology__
> Variability in electrode trajectory, local tissue inflammation, and microenvironment differences can cause certain contacts to exhibit weak, atypical LFP signatures [6], or ambient noise. A functional embedding should deviate from anatomy whenever the recorded signal is noisy or does not reflect typical physiology.\
> __4. Chance level is low and the task is zero-shot__
> With 7 regions, chance performance is 14.3%. Zero-shot classification of unseen channels achieving \~44% (\~50% in aggregated model) indicates strong diagonal dominance and functional clustering despite label noise. Importantly, the goal of the embedding is not to maximize classification accuracy but to learn a continuous, functionally meaningful representation used downstream.\
> __5. Data limitations artificially reduce channel-level accuracy__
> With only 10 seconds per embedding and only 1–3 contacts per region per subject, the model has limited examples to learn fine-grained distinctions. The fact that classification is better for GPi (where we have more contacts), at nearly 70%, supports the hypothesis that performance will improve with additional data. Indeed, our dataset is actively growing.\
> Finally, prior work shows that basal ganglia and thalamic regions exhibit region-specific oscillatory and bursting patterns [7], so using region labels as a supervisory signal is well-justified.\
> Overall, while held-out channel accuracy is a partial evaluation of the learned functional map and, perfect validation might be impossible, the __large downstream performance gains over MNI__ in the transformer demonstrate that the embeddings capture functionally meaningful structure beyond anatomy (at least in the context of inter-regional relationships which is the end task).
> ## (2) Confusion-matrix readability
> We agree and will replace Figures 2–3 with higher-resolution matrices and improved labeling in the revision.
> ## Refs
> [1] Wiesman, Alex I., et al. (2022). "Stability of spectral estimates in resting-state magnetoencephalography: Recommendations for minimal data duration with neuroanatomical specificity."\
> [2] Sillay, Karl A., et al. (2013). "Perioperative brain shift and deep brain stimulating electrode deformation analysis: implications for rigid and non-rigid devices."\
> [3] Williams, Brendan, et al. (2024). "Thalamic nuclei segmentation from T1-weighted MRI: Unifying and benchmarking state-of-the-art methods." \
> [4] Pedrosa, David J., et al. (2018). "A functional micro-electrode mapping of ventral thalamus in essential tremor."\
> [5] Kim, Dae‐Jin, Bumhee Park, et al. (2013). "Functional connectivity‐based identification of subdivisions of the basal ganglia and thalamus using multilevel independent component analysis of resting state fMRI."\
> [6] Rassoulou, Fayed, et al. (2025). "Electrophysiological signatures predict the therapeutic window of deep brain stimulation electrode contacts."\
> [7] Buzsáki, György. (2006). “Rhythms of the Brain.”

---

> ### Author Response · Authors · 2025-11-27
> **Response to questions**
>
> We appreciate the reviewer’s thoughtful questions, and we address each of them in detail below.
> ## Q1 Why do the reconstructed Func1/Func2 signals show higher-frequency content than MNI reconstruction?
> The key reason is __insufficient coordinate resolution under MNI tokenization__.\
> In the illustrated example, pairs of VO channels share identical MNI coordinates (due to voxel discretization and the spatial proximity of neighboring contacts). When the transformer receives identical coordinate tokens for two distinct channels, it is forced to produce similar predictions, effectively averaging their distinct waveforms. This averaging suppresses higher-frequency components and yields smoother, lower-frequency reconstructions.\
> In contrast, functional embeddings are unique for each channel irrespective of physical proximity of electrodes, reflecting channel-specific physiology (e.g., spectral profile, burstiness). As a result, the transformer receives distinct functional tokens for each VO channel and can reconstruct their higher-frequency structure more accurately.\
> There is a step-like attribute present in all the predictions which is the artifact caused by tokenization scheme segmenting signal into 25ms segments.\
> We want to thank you for the insightful question.
> ## Q2 Were the withheld VO channels excluded from functional encoder training?”
> We appreciate this question and agree that differentiating these conditions is important. In the current dataset, three subjects (S2 ,S6 and S11) used in the reconstruction experiments were not included in the functional-embedding training at all, providing preliminary evidence that the pretrained embedding generalizes across subjects.\
> To address this more rigorously, we have added an extended analysis (Appendix A7) where:\
> • n subjects are entirely withheld from functional embedding training,\
> • Masked-region reconstruction is then trained and evaluated on those withheld subject.\
> This directly tests whether functional embedding learned from region labels generalize zero-shot to unseen subjects. We have included these results in the revised paper.\
> Conceptually, once a sufficiently robust functional map is learned from labeled data, the embedding stage functions as a shared feature extractor. New subjects (even those without region labels) need only provide a short (10s) recording to obtain functional embeddings. This substantially expands the applicability of the method to larger unlabeled datasets.

---

### Official Review · Reviewer_rm8d · 2025-11-01

**Soundness:** 2
**Presentation:** 2
**Contribution:** 4
**Rating:** 4
**Confidence:** 5

**Summary:**

The paper proposes FunctionalMap, a contrastive method for learning functional embeddings of sEEG channels over multiple patients (and hence multiple electrode configurations). The method consists of a Siamese convolutional encoder on neural recordings and a contrastive objective where embeddings of channels from the same brain region are pulled together, while those from different regions are pushed apart. The method is evaluated on synthetic data to uncover curated region-specific signatures of oscillatory activity from generated time series, and on a real dataset with brain region decoding and cross-region channel reconstruction as primary tasks. From the experiments, the authors suggest that the proposed method learns channel-level embeddings that can uncover continuous spectral features, spatial information, and results in improved reconstruction performance when compared to selected baselines. The paper argues that data-derived functional coordinates are a more reliable backbone for large-scale intracranial recordings than anatomical alignment.

**Strengths:**

(S1) Clear motivation and relevant work. The reliance on spatial coordinates in previous works modeling sEEG reflects a clear gap in the ability to generalize beyond spatial priors and to datasets where coordinates are unavailable or unreliable. Learning a data-driven, region-structured embedding first, then using it for tokenization downstream is clean, modular, and well-justified for heterogeneous sEEG setups.

(S2) By removing reliance on subject-specific heads or IDs, FunctionalMap enables zero-shot transfer to new subjects without having to lose channel identity. As suggested by the authors, this would be critical for large-scale pretraining on sEEG data, so subsequent works can focus on developing robust methods for representational learning on, or decoding from, neural signals without the constraint of having to learn new identities during transfer.

(S3) The paper presents interesting analyses that highlight properties of the method. In particular, I appreciate the study on synthetic data which demonstrates that the embeddings recover oscillatory features in a continuous space. Further, the comparison and analysis of contrastive objectives is useful for understanding how different formulations affect the geometry of the learned embedding space and downstream performance.

**Weaknesses:**

(W1) The real dataset on which the method was evaluated is limited, especially in terms of spatial coverage. While the isolated study of 5 brain regions is useful for analyzing the method in a controlled setting, the extent of the results is limited without extensions to other datasets. It would be ideal to see how the model fares on a larger dataset with more coverage of brain areas, such as Braintreebank \[1\] which is rich in terms of spatial coverage of sEEG recordings and would create a connection to previous works:

1. To build confidence that FunctionalMap can generalize, it would be nice to see how well it does on brain region decoding / reconstruction tasks on a new dataset, i.e. pretrain on one dataset and transfer to another.
2. For the proposed application of the method on downstream large-scale pretraining / decoding to be viable, it should be demonstrated that FunctionalMap can itself scale up to pretraining on multiple datasets. I would suggest jointly pretraining on multiple datasets that span many brain areas.

(W2) The approach still depends on region labels for the contrastive pretraining, hence the motivation to move away from localization-based labels is not fully addressed. For instance, localization information can be imperfect due to topographic variability (e.g. due to resected areas \[2\]), electrode straddling over time, and since brain region labeling is a process done post-hoc it further invites the possibility of label noise.

(W3) Further, while the method does enable zero-shot transfer at inference-time, it assumes sufficient coverage during pretraining. While there are large datasets available with localization, there are also abundant corpora of data with no spatial coordinate labels. The method is fundamentally limited in its ability to scale up on these datasets which represent a large proportion of the publicly (and often privately) available sEEG data (e.g. \[3\]).

(W4) The paper lacks any sort of scaling analysis that could characterize how best to utilize the method. For instance, it would be useful to see how much data (along axes like \# of subjects, \# of sessions, number of recording hours, number of channels) is necessary to achieve the generalization capabilities highlighted in the results. Such an analysis would inform future studies and clarify the minimum data regime required for stable cross-subject alignment.

(W5) The paper could benefit from clearer framing and additional editing. For instance, sections 4.2 and 4.3 do not introduce the brain region decoding task at all. While it can be inferred from Figures 3 and 4 that the task in question is indeed brain region decoding, it was not set up or specified at all in the sections. There are also a number of minor grammatical and stylistic mistakes throughout the text.

**Questions:**

(Q1) How sensitive is the contrastive method to imperfect region labels, as detailed in (W2)? Could the authors perform a robustness analysis where label noise is artificially introduced in the brain region labels to approximate more realistic clinical localization?

(Q2) To clarify, as suggested at the beginning of section 3.2, were functional embeddings inferred on a new subject on only a single 10s segment, then the rest of the signals were used arbitrarily in the downstream decoding (reconstruction) task? If so, this reveals a very promising property of the zero-shot capability of the model. However, I’m interested in understanding the scope of this a bit more: (1) Does it matter which 10s segment is considered? (2) Does this property hold when generalizing to new brain regions? (3) Would the decoder model benefit from accepting a dynamic embedding, e.g. pass each context window being inputted into the decoder through FunctionalMap to zero-shot derive per-context-window, per-channel functional embeddings?

(Q3) The simulation study is limited only to oscillatory patterns. While these are very commonly strong signals for various stimuli / behaviors coded in the brain, there are other types of patterns that may be relevant. Of course one cannot exhaustively curate all possible patterns, and this isn’t necessarily required, it would be nice to see some diversity. Examples such as transient bursts \[4\] or cross-frequency coupling \[5\] could be considered. How would the model fare in being able to model these types of patterns, and would we observe the same sort of continuous mapping? Also, the study on synthetic data offers a nice opportunity to scale up the number of patients without worrying about data scarcity.

(Q4) Can confusion matrices on brain region decoding and PCA plots from the embedding space be provided for the other subjects, e.g. in the appendix? It’s not clear whether the observed spatial structure in S4’s learned embeddings generalizes to the others.

(Q5) To clarify, sections 4.2 and 4.3 do refer to brain region decoding as a task?

(Q6) What would the authors expect from including subject-specific components (e.g. subject ID/embeddings, decoder head, learnable channel embeddings), i.e. is the line of thinking that FunctionalMap would enable on–par or better downstream performance due to a better channel-level prior? Or do the authors expect performance to degrade, yet the ability to generalize in a zero-shot manner offsets the gap? It would be insightful to analyze this tradeoff.

(Q7) Can the authors comment on the choice to not include projection heads before the contrastive loss? Since this is standard in modern contrastive learning methods \[6\], it would be good to validate this design choice.

Lastly, a quick stylistic suggestion: on Figure 2C, it might be better to color the frequency on a log-scale.

\[1\] Wang, C., Yaari, A., Singh, A., Subramaniam, V., Rosenfarb, D., DeWitt, J., ... & Barbu, A. (2024). Brain treebank: Large-scale intracranial recordings from naturalistic language stimuli. Advances in Neural Information Processing Systems, 37, 96505-96540.

\[2\] Roberts, D. W., Hartov, A., Kennedy, F. E., Miga, M. I., & Paulsen, K. D. (1998). Intraoperative brain shift and deformation: a quantitative analysis of cortical displacement in 28 cases. Neurosurgery, 43(4), 749-758.

\[3\] Carzaniga, F., Hersche, M., Sebastian, A., Schindler, K., & Rahimi, A. (2025). A foundation model with multi-variate parallel attention to generate neuronal activity. arXiv Preprint arXiv:2506. 20354\.

\[4\] van Ede, F., Quinn, A. J., Woolrich, M. W., & Nobre, A. C. (2018). Neural oscillations: sustained rhythms or transient burst-events?. Trends in neurosciences, 41(7), 415-417.

\[5\] Canolty, R. T., & Knight, R. T. (2010). The functional role of cross-frequency coupling. Trends in cognitive sciences, 14(11), 506-515.

\[6\] Gupta, K., Ajanthan, T., Hengel, A. V. D., & Gould, S. (2022). Understanding and improving the role of projection head in self-supervised learning. arXiv preprint arXiv:2212.11491.

---

> ### Author Response · Authors · 2025-12-03
> **response to weaknesses**
>
> We thank the reviewer for the thoughtful feedback. Below we address each concern directly. We thank the reviewer for the thoughtful feedback. Below we address each concern directly.
> ## (W1) “Dataset is limited; unclear if the method generalizes to larger/more diverse datasets.”
> We thank the reviewer for this thoughtful suggestion. While applying FunctionalMap to a whole-brain dataset such as Braintreebank is an exciting future direction, we do not view the current dataset as limited for the scientific and clinical setting this paper targets. The 5–7 regions studied here are not arbitrary: they constitute the core nodes of the basal ganglia–thalamocortical (BGTC) loop, which is central to movement control and is the primary target of deep brain stimulation in clinical practice [1]. Recordings from GPi, STN, VO/VIM, and related nuclei are increasingly common, and modeling their interactions is a highly active area of neuroscience and neurotechnology [2].\
> In this domain, cross-subject alignment of deep-brain recordings is extremely valuable. FunctionalMap provides, to our knowledge, the first functional coordinate system that enables such alignment, and our results show that aggregating 20 subjects already yields substantially better performance than training on a single subject. This demonstrates the core contribution: establishing a framework that enables cross-subject modeling in deep nuclei, where anatomical variability is high [3] and functional relationships are critical.\
> We agree that scaling to larger datasets and whole-brain coverage is an important next step. However, BGTC modeling alone is a major application domain, not a limited one: deep nuclei are densely sampled in clinical datasets, and understanding their circuit dynamics and involvement in movement disorders is a high-priority research direction. Future work can extend FunctionalMap to broader coverage, but within the BGTC circuit, the method is already both appropriate and impactful.\
> Finally, to address the reviewer’s suggestion directly, we have performed a new scalability analysis, results show increase in performance with increase in subjects has an early plateau at 8 but keeps increasing after 15 subjects. Results reported in appendix A.10 in the revised paper.
> ## (W2) “Method depends on region labels; motivation to move away from localization-based labels not fully addressed.”
> We thank the reviewer for raising this important point. In fact, the concern they highlight is one of the core motivations for our approach. While anatomical labels in deep nuclei are imperfect due to brain shift, indistinct borders, and post-hoc reconstruction variability, we assume that even noisy labels contain enough shared structure for the model to extract region-relevant physiological features across subjects.\
> Two clarifications are essential:\
> (1) Region labels are only required once, during the initial encoder training.\
> After this stage, the functional encoder operates as a shared feature extractor and can be applied to new datasets without region labels. This addresses the reviewer’s concern directly: the method enables large unlabeled sEEG corpora to be incorporated into downstream modeling. To demonstrate this empirically, we added an analysis (Appendix A.7) where five subjects were held out entirely from pretraining, and we observed no degradation in downstream masked-region reconstruction.\
> (2) Label noise does not undermine the approach, rather, it motivates it.\
> Because the encoder receives only LFP waveforms (no subject or session IDs), the only way it can solve the contrastive task is by identifying neural features that are consistent across subjects within a region and different across regions. Even under imperfect labels, our results show that such features do exist and support a stable data-driven coordinate system.\
> Together, these points demonstrate that while region labels seed the initial training, FunctionalMap ultimately provides a label-free, physiology-based coordinate system that improves downstream performance and overcomes the very limitations of anatomical localization highlighted by the reviewer.
> ## Refs
> [1] Vitek, J. L. (2008). "Deep brain stimulation: how does it work?"\
> [2] Shaheen, H., & Melnik, R. (2022). "Deep Brain Stimulation with a Computational Model for the Cortex‐Thalamus‐Basal‐Ganglia System and Network Dynamics of Neurological Disorders."\
> [3] Segobin, Shailendra, et al. (2024). "A roadmap towards standardized neuroimaging approaches for human thalamic nuclei."

---

> ### Author Response · Authors · 2025-12-03
> **response to weaknesses (continue)**
>
> ## (W3)“Method assumes sufficient labeled coverage; cannot scale to unlabeled datasets.”
> We appreciate the reviewer raising this point. In fact, this concern directly motivates the development of our data-driven coordinate system. While many sEEG datasets lack reliable anatomical coordinates or region labels, FunctionalMap is specifically designed to overcome this limitation, not be constrained by it.\
> Two points address the reviewer’s concern:\
> (1) Region labels are only needed once, during initial pretraining.\
> After a functional encoder is trained on a modest labeled dataset, it can be applied to any unlabeled dataset. Our held-out subject analysis (five subjects excluded from pretraining) shows no reduction in downstream performance, demonstrating that once the functional map is learned, new subjects do not require labels. This has been explicitly shown in the revised paper appendix A.7 and is precisely what enables scaling to large unlabeled corpora.\
> (2) When datasets lack MNI coordinates, as is common with small-depth electrodes, MNI-based modeling becomes impossible.\
> In these cases, our method provides the only viable way to train cross-subject models. Anatomical-coordinate approaches fundamentally cannot scale to unlabeled datasets, whereas FunctionalMap produces label-free embeddings that generalize across subjects.\
> Thus, rather than being limited by unlabeled datasets, the proposed approach enables access to them, which is a key advantage over coordinate-based methods. We will emphasize this explicitly in the revised manuscript.
> ## (W4) “Paper lacks scaling analysis; unclear how much data is required for generalization.”
> We thank the reviewer for this insightful suggestion. We agree that understanding how performance scales with dataset size is important for guiding future applications. In response, we have performed a new scalability analysis Finally, to address the reviewer’s suggestion directly, we have performed a new scalability analysis, results show increase in performance with increase in subjects has an early plateau at 8 but keeps increasing after 15 subjects. Results reported in appendix A.10 in the revised paper.
>  ## (W5) paper formatting
> We appreciate this feedback and will address these issues in the revised manuscript.

---

> ### Author Response · Authors · 2025-12-03
> **response to the questions**
>
> ## (Q1) “Sensitivity to imperfect region labels; robustness to label noise.”
> This is an excellent question. In practice, region labels already contain substantial noise, and the contrastive method is designed to function under exactly these conditions. Importantly, label noise comes from two distinct sources:\
> (1) Anatomical noise:\
> Errors in MRI localization, inter-subject variability, brain shift, and indistinct thalamic boundaries can cause mislabeling even when the electrode is physically in the intended nucleus [1,2].\
> (2) Functional noise:\
> Poor tissue–electrode coupling, inflammation, or individual differences in physiology can cause electrodes with the same anatomical label to exhibit very different LFP features.\
> These two noise sources have different implications:\
> •	Anatomical noise can be partially mitigated by excluding borderline electrodes.\
> •	Functional noise cannot, and should not, be removed, because the goal of the functional embedding is precisely to separate electrodes whose physiology does not match their anatomical label. This is why perfect label recovery is neither expected nor desirable.\
> ## (Q3)“Simulation includes only oscillatory patterns; what about bursts or CFC?”
> We agree with the reviewer that including a variety of signal motifs is important. In fact, this is already incorporated in our simulation. Appendix A.2 introduces burst events as region-specific features, and Figure 2B shows through a perturbation study that the network correctly localizes these bursts within the 10-second windows and uses them to identify the corresponding region. Thus, the synthetic data do include more than sustained oscillations, and the model successfully captures these transient dynamics.
> ## (Q4) single subject plots
> Yes, we have added the confusion matrices and PCA/embedding plots for all subjects in Appendix A.11 of the revised manuscript. These confirm that the spatial structure observed in S4 generalizes consistently across the full cohort.
> ## (Q5) “Do Sections 4.2 and 4.3 refer to brain region decoding as a task?”
> Yes. These sections evaluate the brain region decoding task, but we clarify that this analysis is not intended as a standalone downstream task. Rather, it is used to assess whether the learned functional coordinate system captures shared structure across subjects, which in turn enables the downstream masked-region reconstruction. It can, however, also be viewed as a secondary downstream task.
> ## (Q6) “What if subject-specific components (IDs, embeddings, learnable channel tokens) were added? Would performance improve or degrade?”
> This is an interesting point. However, adding subject-specific information is counter to the primary goal of our work. In BCI-style applications, optimizing performance per subject is desirable, and models often explicitly encode subject identity. In contrast, our objective is to learn subject-agnostic features that reflect shared physiology of the basal ganglia-thalamic circuit. Introducing subject IDs or subject-specific embeddings would encourage the model to rely on idiosyncratic cues (e.g., noise structure, lead-specific artifacts, 60 Hz contamination), rather than the cross-subject features that reflect true inter-regional connectivity.\
> As a result, such components would likely improve within-subject reconstruction but degrade zero-shot generalization and obscure the shared dynamics we aim to model. For these reasons, we deliberately avoid subject-specific elements in FunctionalMap.
>
> ## (Q7) “Why no projection head before the contrastive loss, given its common use?”
> We thank the reviewer for raising this point. While projection heads are indeed standard in many modern contrastive learning frameworks, we chose not to include one for two reasons. First, we aimed to keep the embedding space directly interpretable, since part of our analysis examines which physiological features the encoder relies on; adding a non-linear projection head would obscure this. Second, our results show that the method already performs well without a projection head. As in other contrastive-learning settings, one can always discard the final layer for downstream tasks, but in our case, the simpler architecture aligns better with our interpretability goals.
>
> ## Final style comment
> Thank you for the suggestion. We experimented with a log-scale color mapping, but found that it reduced visual clarity. For this reason, we kept the linear scale in the final figure.
>
> ## Refs:
> [1] Halpern, et al. (2007). "Brain shift during deep brain stimulation surgery for Parkinson’s disease."\
> [2] Williams, et al. (2024). "Thalamic nuclei segmentation from T1-weighted MRI: Unifying and benchmarking state-of-the-art methods."

---

> ### Author Response · Authors · 2025-12-03
> **respones to question2**
>
> ## (Q2): “Are embeddings for a new subject inferred from a single 10 s segment, and how broad is this zero-shot capability?”
> Yes, as described in Section 3.2, embeddings for each channel in a new subject are computed from a single 10-second segment at the beginning of the recording session, and those fixed embeddings are used throughout the downstream masked-region reconstruction. This is indeed a key aspect of the zero-shot generalization capability.\
> We clarify each sub-question below:\
> (1) Does it matter which 10 s segment is used?\
> In practice, we did not observe meaningful differences when selecting different 10-second windows within the same session. LFP spectral structure in these deep nuclei is relatively stable over short time scales, so any short window provides a representative sample for functional embedding. For consistency, all embeddings were extracted from the first 10 s of each session.\
> (2) Does this property hold when generalizing to new brain regions?\
> Yes, the embedding is computed per-channel, regardless of region, and applies equally to all regions present in the recording. In the transformer experiments, channels from VO, the masked region, were not included in functional-encoder training for several subjects (e.g., S2, S6, S11), yet their 10-second embeddings still improved downstream reconstruction. This indicates that the zero-shot property extends to channels from regions not seen during functional-encoder training. an additional more systematic analysis has been added as mentioned before for heldout-subject study Appendix (A.7)\
> (3) Would a dynamic (per-window) embedding help?\
> We designed the encoder to produce time-stable, channel-level embeddings, not dynamic state embeddings. This choice aligns with the goal of learning a functional identity for each channel (analogous to an anatomical coordinate), which should remain constant across the session. A dynamic embedding could encode transient state changes rather than stable inter-regional relationships and might therefore confound the transformer’s ability to learn subject-agnostic structure.\
> However, exploring dynamic embeddings is an interesting future direction, and we will note this in the revised manuscript.

---

### Official Review · Reviewer_KLms · 2025-11-05

**Soundness:** 3
**Presentation:** 2
**Contribution:** 3
**Rating:** 6
**Confidence:** 4

**Summary:**

This paper addresses the challenge of aggregating heterogeneous, multi-subject intracranial (SEEG) recordings.  This task is notoriously complicated due to variable electrode placements.  The authors propose a framework that learns a subject-agnostic functional identity for each electrode, effectively creating a learned "functional coordinate system".   To my knowledge, this appears to be the first time such a learned functional coordinate system was attempted for Ephys data.  The training approach uses contrastive learning with annotated brain functions in the training data.  Afterwards, this learned functional embedding (called functional tokens) was fed into downstream "functional transformer".  The validation dataset is a 20-subject dataset, showing improved performance on models that rely on more rigid coordinate systems.

**Strengths:**

The paper's primary strength lies in its novel approach (functional tokens) for dealing with unreliable anatomical coordinates in Ephys data.  The empirical evidence supports the benefits of this approach, including both analyzing whether the functional coordinate system is correlated with ground truth functional annotations, as well as the benefits of it for downstream decoding.

**Weaknesses:**

I think the paper over-weights interpreting the functional coordinate system (e.g., all the various confusion/correlation analyses), and under-weights understanding how this embedding improves downstream decoding performance.

The main result for downstream decoding is Figure 6, which I found difficult to interpret.  Moreover, I would've liked to see more analysis teasing apart how and when the functional tokens improve decoding performance.

**Questions:**

No additional questions, beyond commenting on my weakness items.

---

> ### Author Response · Authors · 2025-11-26
>
> We thank the reviewer for the thoughtful feedback. Below we address each concern directly.
> # (1) “The paper overweights interpreting the functional coordinate system”
> We appreciate this observation. In the revision, we will move additional exploratory analysis of the embedding (confusion matrices, correlation analyses of single subjects) into the Appendix.\
> We also clarify why the contrastive-learning component received substantial emphasis in the initial draft. To our knowledge, this is the first data-driven functional coordinate system for intracranial LFPs from basal ganglia, and it was not obvious a priori that stable, region-specific signatures in the 0–50 Hz band exist across subjects (despite some evidence of region specific spectral properties that made this hypothesis reasonable [1,2]). Establishing presence of these signitures, and that a continuous functional map can be learned, parallels classical efforts such as the Penfield cortical homunculus [3], where functional organization was inferred from physiological responses rather than anatomy. Here, functional embeddings similarly act as an empirically derived “map” of deep brain structures, overcoming well-known limitations of anatomical coordinates (e.g., indistinct borders and inter-subject variability) [4,5].\
> This is important clinically: distinguishing thalamic nuclei from imaging alone is unresolved [4,6], and functional mapping provides complementary evidence.
> # (2) “Figure 6 is difficult to interpret; more analysis is needed to understand when functional tokens improve decoding.”
> Thank you for pointing this out. We will clarify Figure 6 and add additional downstream analyses.
> ## Why masked-region reconstruction is an appropriate downstream task
> Our goal is to evaluate models in a way that (a) is shared across subjects, (b) relies only on signals within the basal ganglia (BG) circuit, and (c) is challenging enough that cross-subject aggregation provides a tangible benefit.\
> Masked-region reconstruction satisfies all three conditions:\
> •	Shared structure across subjects:\
> Inter-regional dependencies (e.g., GPi to VO dynamics) reflect conserved physiology across individuals.\
> •	Circuit-level predictability:\
> Unlike behavior decoding, which depends on many unobserved pathways (cortex, spinal cord) and varies strongly across individuals, predicting activity in a thalamic nucleus (e.g., VO) from GPi/STN/VIM relies on interactions within the BG loop, where we have simultaneous recordings. Prior work shows that basal ganglia LFPs are substantially less behavior-predictive than cortical signals [7], so behavior decoding is not an ideal shared downstream target for this dataset.\
> •	Task difficulty:\
> A single subject typically has only 1-3 recordings per region, which is insufficient for learning inter-regional dependencies. Aggregating across subjects provides enough coverage to learn meaningful structure.
> ## Impact and relevance
> Unlike cortical brain computer interface work, basal ganglia LFPs are rarely used for behavior decoding. In this domain, inferring the activity of an unrecorded area from other nodes in the basal ganglia loop is itself a highly desirable scientific and clinical end-task, not merely a pretraining proxy. Many clinically relevant biomarkers (e.g., STN β-power in Parkinson’s disease [8]) depend on interactions across BG regions, and being able to reconstruct VO activity from GPi/STN/VIM provides a direct tool for studying these circuit relationships, and enabling “virtual sensing’’ of nuclei that cannot be implanted.
> # To address this concern,
> we added a band–power prediction task (Appendix A.8). Band–power is a clinically meaningful and interpretable measure of neural oscillations, so predicting it from distributed activity provides a practical downstream test of our functional coordinates. The revised manuscript includes these results and shows that functional tokens outperform MNI tokens on this task.
> # Refs
> [1] Bush, Alan, et al. (2024). "Aperiodic components of local field potentials reflect inherent differences between cortical and subcortical activity.”\
> [2] Buzsáki, György. (2006). “Rhythms of the Brain.”\
> [3] Penfield, Wilder, et al. (1937). "Somatic motor and sensory representation in the cerebral cortex of man as studied by electrical stimulation."\
> [4] Williams, Brendan, et al. (2024). "Thalamic nuclei segmentation from T1-weighted MRI: Unifying and benchmarking state-of-the-art methods." \
> [5] Hoch, Michael J., et al. (2022). "MRI-visible anatomy of the basal ganglia and thalamus."\
> [6] Segobin, Shailendra, et al. (2024). "A roadmap towards standardized neuroimaging approaches for human thalamic nuclei."\
> [7] Merk, Timon, et al. (2022). "Electrocorticography is superior to subthalamic local field potentials for movement decoding in Parkinson’s disease."\
> [8] Neumann, Wolf‐Julian, et al. (2016). "Subthalamic synchronized oscillatory activity correlates with motor impairment in patients with Parkinson's disease."

---

### Author Response · Authors · 2025-12-03
**Message to Area Chair**

We thank the reviewers and the committee for their careful consideration and constructive feedback. Here we briefly clarify the intended scope of the work, address what we believe are a few key misunderstandings, and summarize the revisions we have made.
### 1. Scope and novelty of the contribution
Our primary goal is circuit-level modeling of basal ganglia–thalamic (BGTC) activity across subjects, not cortical BCI-style behavioral decoding. In this domain:\
•	Predicting neural activity in an unrecorded deep region from other BGTC nodes is itself a core scientific and clinical end-task, not merely a pretraining objective.\
•	To our knowledge, this is the first data-driven functional coordinate system for intracranial LFPs in human basal ganglia and thalamus, and the first large-scale cross-subject modeling of these nuclei using such a functional map.\
Because anatomical coordinates in deep structures are noisy and often unavailable, existing cross-subject methods cannot be applied to many large sEEG datasets. Once our functional encoder is trained on a modest labeled cohort, it can be applied zero-shot to new subjects and to datasets without anatomical coordinates, enabling cross-subject modeling precisely in the regime where no alternative currently exists. We now make this point explicit in the revision.
### 2. On “triviality” of supervised contrastive learning and held-out channel accuracy
Some critiques treat the region-separating embedding as an almost “expected” outcome of supervision. However, it was not known a priori that 10 s, 0–50 Hz LFP segments from deep nuclei would contain stable, region-specific signatures across subjects strong enough to support a shared functional map. Demonstrating that such a map exists and that it can be learned and transferred across subjects, is already a nontrivial neuroscientific result.\
Similarly, concerns about the “low” held-out channel accuracy do not fully reflect our objectives:\
100% alignment with anatomical labels is neither expected nor desirable. Deep-brain labels are noisy proxies (brain shift, indistinct thalamic borders, poor contacts); a functional embedding that partially disagrees with anatomy, especially near boundaries and in noisy channels, is precisely what can improve on MNI in downstream tasks.\
Crucially, regardless of these intermediate metrics, we show that replacing MNI coordinates with functional tokens consistently improves downstream performance, and we now demonstrate this on two downstream tasks.
### 3. Masked reconstruction is not “just pretraining”
Several comments implicitly compare our setup to self-supervised cortical BCI pipelines, where masked reconstruction is used only as pretraining for behavior. In our setting:
•	Basal ganglia LFPs are rarely used for high-fidelity behavioral decoding; they are primarily studied to understand BGTC circuit dynamics and clinical biomarkers.\
•	Masked-region reconstruction of VO from GPi/STN/VIM is itself a meaningful endpoint: it enables “virtual sensing” of nuclei that cannot be implanted and directly tests whether the model has learned stable inter-regional dependencies shared across subjects.\
We believe this mismatch in expectations (BCI framing vs BGTC circuit modeling) underlies some of the skepticism about our evaluation.
### 4. Revisions made in response to reviewer feedback
In direct response to the reviews, we have:\
•	Added a clinically relevant power-band prediction task as a second downstream evaluation (Appendix A.8), showing that functional tokens improve performance beyond MNI in an additional, interpretable task. Band-power is a clinically meaningful, interpretable measure of neural oscillations (e.g., alpha, beta, gamma). Accurately predicting band-power from distributed activity provides a practical downstream task that directly tests the utility of our learned functional coordinates.\
•	Added a systematic held-out subject analysis (Appendix A.7), demonstrating that subjects excluded from encoder training still benefit from the functional map in downstream tasks, reinforcing zero-shot generalization and scalability to unlabeled datasets.\
•	Added a scalability analysis for the effect of aggregation of subjects in the downstream task in Appendix A.10. With maximin performance using all subjects, yet seeing gains as early as 5-8 subjects.\
•	Added a region-tag baseline (Appendix A.9) showing that learned functional embeddings outperform simple region-label or one-hot embeddings, confirming that gains are not due to label inclusion alone.\
•	Added all single subject results and embeddings (Appendix A.11).\
Overall, we believe the revised paper more clearly communicates that this work is not an incremental variant of BCI pretraining, but rather a foundational tool toward subject-agnostic, data-driven cross-session modeling of brain circuits, with immediate relevance for large-scale neuroscience and clinical research.

---

### Meta-Review · Area_Chair_CiMZ · 2026-01-06

**Summary:**

The paper tackles the problem of aggregating heterogeneous multi-subject intracranial sEEG/field-potential recordings, where analysis is difficult because electrode placements vary substantially across patients. To address this, the authors propose FunctionalMap, a contrastive framework that learns subject- and region-agnostic functional embeddings for individual channels/electrodes—effectively a learned “functional coordinate system” (functional tokens) intended to serve as a more reliable backbone than rigid anatomical alignment (e.g., MNI).

Methodologically, the approach uses a Siamese/contrastive encoder (implemented as a pairwise Siamese contrastive objective and a modified supervised-contrastive/SupCon variant) that pulls together embeddings from channels labeled as the same brain region and pushes apart embeddings from different regions. The learned embeddings are evaluated both on synthetic data, where they recover curated region-specific oscillatory signatures and enable near-perfect region classification, and on real multi-subject data, where they support brain-region decoding under different cross-validation schemes (e.g., held-out time segments vs. held-out channels) and improve performance when training across subjects (reported as a modest accuracy gain).

Finally, the learned embeddings are used as functional tokens for a downstream transformer-based reconstruction model, where functional tokenization is reported to outperform anatomical coordinate tokenization (e.g., MNI), with the modified SupCon variant generally preferred over the pairwise Siamese setup. Overall, the paper argues that data-derived functional coordinates provide a better way to align and model large-scale intracranial recordings across subjects than relying on fixed anatomical coordinate systems.

**Reviewer Concerns:**

A consolidated list of the main reviewer concerns (ordered by importance in my view) is provided below:

* **Downstream impact not established (and evaluation is indirect).** It remains unclear whether the learned “functional coordinates/tokens” *actually* advance the stated goal of cross-subject pretraining and decoding. Reviewers found the main downstream evidence hard to interpret, and noted that improved reconstruction loss does not necessarily translate to better decoding or behavioral prediction. They wanted clearer analyses showing *when/how* the embeddings help and direct downstream gains using the pretrained functional transformer.

* **Generalization across channels/subjects is questionable.** The evidence for transferable functional alignment in real data is mixed: performance on held-out channels is substantially lower than on held-out time segments (even if above chance), suggesting limited channel-level invariance under the current data regime (~6–12 channels/region). Moreover, cross-subject aggregation appears to provide the largest gains for sessions with low single-subject performance, and it is unclear whether it consistently improves strong sessions or partly trades them off. A leave-subjects-out evaluation and/or downstream cross-subject decoding results would more directly validate subject-agnostic transfer.

* **Supervision/motivation mismatch: reliance on region labels.** Since contrastive pretraining depends on brain-region labels, reviewers argued the method mainly learns to discriminate labeled regions—an expected outcome given the setup—without fully disentangling that this yields functional alignment. They also highlighted that region labels can be noisy/imprecise and may not map cleanly to functional properties, undermining the motivation to move beyond localization-based alignment.

* **Limited breadth and missing scaling/transfer evidence.** The real-data evaluation is limited in spatial coverage, making broad claims hard to support. Reviewers asked for stronger evidence of scalability and generality: larger/more diverse datasets, dataset-to-dataset transfer (pretrain on one, test on another), joint pretraining across datasets, and scaling analyses showing how performance depends on subjects/sessions/hours/channels.

**Reviewer Scores:**

The reviewers’ scores were mixed, but the overall sentiment leaned toward rejection.

One reviewer rated the paper marginally above the acceptance threshold, two rated it marginally below, and one recommended rejection.

While the paper has merit, I believe it would benefit from a major revision of the experimental setup to directly test its core hypothesis—namely, that the proposed functional embeddings provide a transferable, subject-agnostic alignment that improves downstream cross-subject pretraining and decoding, beyond simply producing region-discriminative representations under region-label supervision.

---

### Decision · Program_Chairs · 2026-01-26

Reject